# USDPNET: AN UNSUPERVISED SYMMETRIC DEEP FRAMEWORK FOR ROBUST PARCELLATION OF INFANT SUBCORTICAL NUCLEI

## ABSTRACT

Accurate infant subcortical parcellation is vital for understanding early brain development and neurodevelopmental pathology. However, existing methods suffer from initialization sensitivity, poor bilateral consistency, and limited applicability to early postnatal data. We propose USDPnet, an Unsupervised Symmetric Deep Parcellation Network, which integrates deep autoencoder-based feature embedding with divergence-driven clustering. By introducing the generalized Cauchy-Schwarz divergence (GCSD) as the clustering objective, we enhance inter-cluster separability across complex developmental features. A symmetry constraint further enforces bilateral consistency, leading to structurally coherent and reproducible delineations. USDPnet operates on surface-based features extracted from infant subcortical nuclei. Experiments show it outperforms traditional and deep clustering baselines. Visualizations are largely consistent with the parcellation results based on anatomy and function connectivity. The resulting parcellations are developmentally grounded, anatomically symmetric, and functionally relevant, offering fine-grained and morphological coherent maps of early subcortical organization. Code is available at https://anonymous.4open.science/r/USDPnet-X12D.

## 1 INTRODUCTION

Subcortical nuclei, including the hippocampus, amygdala, thalamus, and basal ganglia (e.g., caudate, putamen, pallidum), form complex neural circuits fundamental to higher-order cognitive functions such as memory consolidation, emotional regulation, and motor control (Johnson, 2012). These structures can be further parcellated into anatomically and functionally heterogeneous subregions (e.g., hippocampal subfields, amygdaloid nuclei) (Tian et al., 2020; Iglesias et al., 2015), each playing distinct roles in behavior and cognition. For instance, CA3a and CA3b subfields of the hippocampus are crucial for encoding spatial memory (Kesner, 2007), while the laterobasal group of the amygdala exhibits robust auditory responses (Ball et al., 2007).

Owing to their specialized functions, subcortical regions are critically implicated in a range of neurological and psychiatric conditions. For instance, atrophy of the CA1 subregion of the hippocampus is a hallmark of Alzheimer's disease (De Flores et al., 2015), while abnormal enlargement of the laterobasal subregion of the amygdala is frequently observed in autism spectrum disorder (ASD) (Kim et al., 2010). Consequently, accurate subregional parcellation of subcortical structures is essential for understanding their underlying mechanisms and informing targeted interventions. Deep brain stimulation (DBS), for example, relies on precise localization of therapeutic targets in nuclei such as the subthalamic nucleus or pallidum (Iorio-Morin et al., 2020).

Recent trends in neuroscience and neuroengineering further emphasize the urgency of high-resolution subcortical mapping. Multimodal neuroimaging atlases (Chen et al., 2022; Tian et al., 2020; Iglesias et al., 2015) and emerging brain-computer interface (BCI) technologies increasingly require anatomically faithful and developmentally informed subcortical boundaries to support fine-grained localization, neural decoding, and closed-loop modulation (Horn et al., 2017).

Notably, early infancy marks a critical period of subcortical growth that underpins the emergence of cognition, consciousness, and neurodevelopmental trajectories (Chen et al., 2023a). Despite this developmental significance, current subcortical parcellation methods fall short in the infant con-

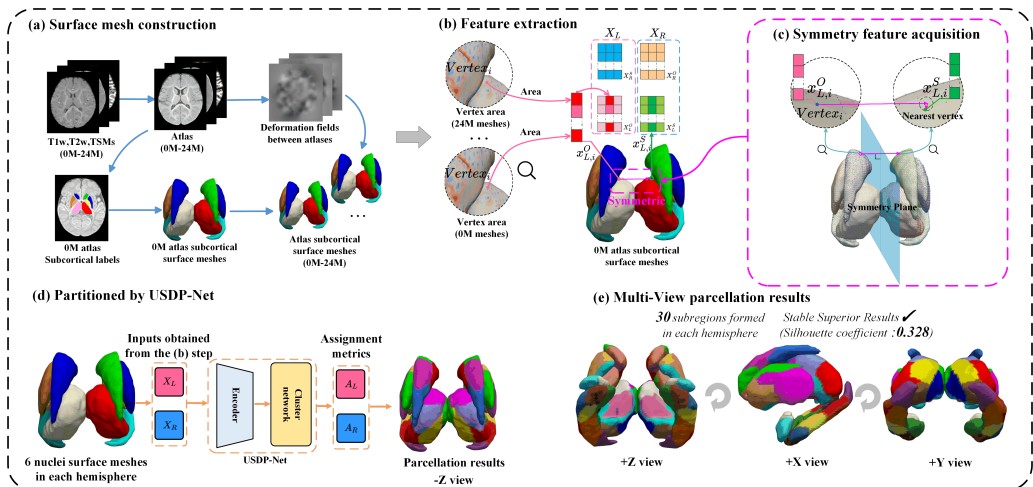

Figure 1: Overview of unsupervised symmetric infant subcortical parcellation. (a) Surface mesh reconstruction of subcortical nuclei across time points within the first two postnatal years; (b) Extraction and integration of vertex-wise area features: original features $X_L^O$, $X_R^O$ and symmetric features $X_L^S$, $X_R^S$ form bilateral datasets $X_L$, $X_R$, where $x_{L,i}^O$ and $x_{R,i}^S$ denote a vertex and its symmetric counterpart; (c) Bilateral correspondence mapping; (d) Architecture of USDPnet; (e) Multi-view visualization of parcellation outcomes.

text. Existing methods are predominantly adult-centric and rely on either static anatomical priors or functional connectivity maps, both of which are difficult to acquire or adapt for infants (Dubois et al., 2021). Compounding this, infant MRIs suffer from motion artifacts, reduced tissue contrast, and rapid intensity fluctuations due to ongoing myelination (Zöllei et al., 2020), rendering most adult-based models ill-suited for infant applications. While recent work has exploited the intrinsic bilateral symmetry of human anatomy (Wathore & Gorthi, 2024; Raina et al., 2020), many existing methods for subcortical parcellation overlook this property (Teyler & Discenna, 1984; Zuccoli et al., 2015). This oversight can lead to mismatched left-right partitions, compromising accuracy and interpretability, particularly in infants where small asymmetries may yield misleading conclusions.

To overcome the limitations of existing subcortical parcellation methods in early infancy, we propose USDPnet—an **U**nsupervised **S**ymmetric **D**eep **P**arcellation Network—which integrates deep autoencoder-based feature embedding with divergence-driven clustering (Figure 1(d)). At the core of our approach, we introduce a novel symmetry-aware clustering architecture specifically designed for bilaterally organized subcortical structures. Thereby, we introduced the generalized Cauchy-Schwarz divergence (GCSD) for supervising each hemisphere to effectively handle complex multi-modal distributions in developmental data, significantly outperforming traditional pairwise measures in multi-region clustering scenarios while substantially improving computational efficiency and stability. To ensure morphologically consistent and interpretable outputs, USDPnet explicitly incorporates bilateral symmetry constraints, aligning with the known anatomical symmetry of subcortical structures across hemispheres. This design not only improves robustness but also ensures consistency of parcellation results. Using 513 high resolution longitudinal infant MRI scans from 231 BCP subjects aged 0-2 years, we construct temporally aligned surface meshes for 6 subcortical structures with infant-dedicated methods and extract vertex-wise area features along with symmetry-consistent correspondence mapping (Figure 1(a-c,e)).

Comprehensive evaluations demonstrate that USDPnet significantly outperforms state-of-the-art methods in clustering quality, anatomical consistency, and computational stability. The resulting subregion maps reveal consistent and interpretable developmental trajectories, offering new insights into early brain organization and subcortical topography. Our main contributions include:

- **Unsupervised and symmetry-aware parcellation framework:** USDPnet enables fine-grained characterization of subcortical nuclei, providing a crucial reference for both developmental neuroscience and translational applications.

- **Robust clustering objective:** We introduce GCSD as a scalable, information-theoretic loss that robustly manages multi-distribution clustering scenarios in high-dimensional feature space.

- **Symmetry-constrained optimization:** By leveraging intrinsic anatomical symmetry, we improve parcellation accuracy, consistency, and interpretability, offering an effective new direction for surface-based neuroimaging parcellation.

## 2 RELATED WORK

**Brain Tissue Parcellation Algorithms.** Parcellation of brain tissue is a fundamental task for understanding the structural-functional architecture of the brain and its role in neuropsychiatric conditions. Glasser et al. (2016) introduced a semi-automated, gradient-based method using multimodal data, achieving 96.6% detection accuracy across 180 cortical areas per hemisphere. In subcortical studies, Tian et al. (2020) used functional connectivity gradients to generate hierarchical atlases, while Jitsuishi & Yamaguchi (2025) applied tractography to map thalamic nuclei to large-scale networks. Cucurull et al. (2018) reformulated Broca's area parcellation as a graph node classification problem, where GCN and GAT outperformed traditional vertex-wise models. Recently, Gao et al. (2025) used diffusion MRI tractography with graph neural networks (GNNs) for fine-scale striatal parcellation. Despite these advances, most methods target adult brains and lack the adaptability to infant developmental characteristics and imaging constraints, calling for tailored solutions for early postnatal parcellation.

**Unsupervised Clustering Methods.** Unsupervised clustering is widely used in parcellation to discover latent structures in neuroimaging data. Classical methods like spectral clustering (Ng et al., 2001) and non-negative matrix factorization (NMF; Lee & Seung, 1999) can model complex and interpretable patterns. Building upon these foundations, deep learning-based unsupervised clustering has emerged as a transformative research paradigm. Deep Embedded Clustering (DEC; Xie et al., 2016) and its extensions such as BDEC(Ma et al., 2023; Zhu et al., 2025) jointly learn representations and clustering in fMRI parcellation tasks.Recent work such as DECS, DCSS, and DMSC(Cheng et al., 2024; Sadeghi & Armanfard, 2023; Zhu et al., 2025) has also improved the mechanisms for balancing the clustering and reconstruction losses in deep clustering. However, these methods suffer from inefficiency in multi-region applications. Clustering effectiveness fundamentally depends on divergence measure quality. Traditional divergences such as Cauchy-Schwarz (Kampffmeyer et al., 2019) or KL divergence (Zhou et al., 2015) are limited to pairwise distributions, and their multi-distribution extensions (Rosenblatt, 2011; Perez, 1984) lack computational scalability. Recently, we proposed a generalized Cauchy-Schwarz divergence (GCSD; Lu et al., 2025b) to address these limitations with efficient and robust multi-distribution estimation, motivating its adoption in USDPnet.

## 3 METHOD

The proposed USDPnet is an end-to-end unsupervised learning framework that jointly performs feature representation learning and clustering-based subcortical parcellation. As illustrated in Figure 2, the model is explicitly designed to maximize inter-cluster separability through a GCSD objective, which is well-suited for capturing complex multi-modal distributions in high-dimensional developmental brain data. At the same time, USDPnet introduces a novel bilateral symmetry regularization to ensure anatomical consistency across hemispheres—particularly critical for the bilaterally organized subcortical nuclei in infant brains. This joint optimization strategy enhances the bilateral consistency, robustness, and interpretability of the resulting parcellations, making USDPnet a principled solution for data-efficient, structure-aware infant brain mapping.

### 3.1 PROBLEM FORMULATION

Accurate and interpretable parcellation of subregions within infant subcortical nuclei is critical for understanding early neurodevelopmental trajectories and the pathophysiology of related neurological and psychiatric disorders. Given the fine-grained and heterogeneous nature of these structures, our goal is to identify developmentally coherent subregions based on local morphological features extracted from surface representations of subcortical nuclei.

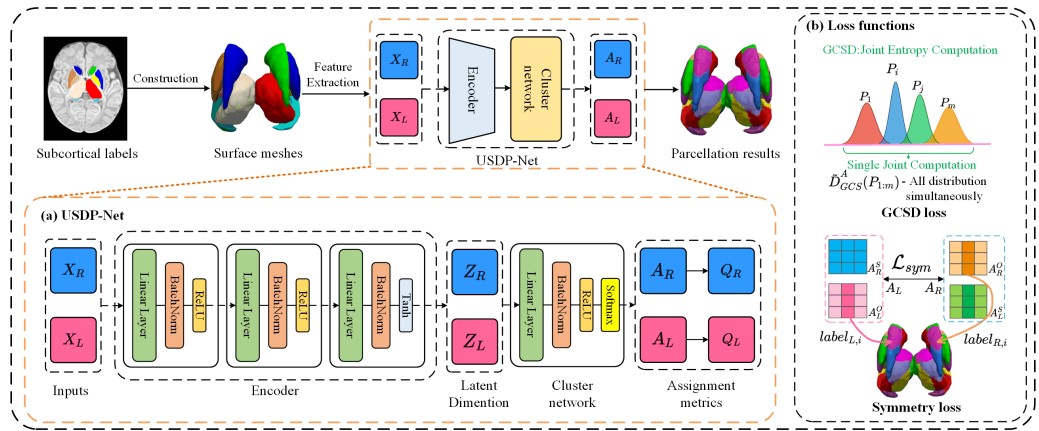

Figure 2: Overall architecture of USDPnet. (a) Bilateral input matrices $X_L$ and $X_R$, containing original and symmetrically mapped surface-area features, are encoded into latent embeddings $Z_L$ and $Z_R$ and then transformed into soft assignment matrices $A_L$ and $A_R$; proximity matrices $Q_L$ and $Q_R$ are further computed from these assignments to characterize cluster-wise similarities. (b) A GCSD-based clustering objective operates on these assignments, where $A_L^O$ and $A_R^O$ denote cluster probabilities for original vertices and $A_L^S$ and $A_R^S$ for symmetrically mapped vertices, and a symmetry loss enforces consistent parcellations across hemispheres.

Formally, let $\{x_i \in X\}_{i=1}^n$ denote a set of high-dimensional feature vectors, where each $x_i \in \mathbb{R}^m$ corresponds to a vertex on the surface of a subcortical nucleus, capturing local developmental metrics (e.g., vertex-wise area over time). The objective is to assign each vertex to one of $r$ anatomically and functionally meaningful clusters $C_1, \dots, C_r$, such that: (1) intra-cluster similarity is maximized; (2) inter-cluster separability is maximized; and (3) anatomical bilateral symmetry is preserved.

This leads to a constrained clustering formulation with dual goals: (1) learning a compact and discriminative latent representation space for subcortical features, and (2) optimizing an anatomically informed cluster assignment that respects the symmetrical nature of infant subcortical structures.

## 3.2 NETWORK ARCHITECTURE

**Overall Architecture.** This study aims to train a network that effectively parcellates each vertex of subcortical nuclei surface models into one of $k$ anatomically consistent subregions. To achieve this, we adopt an encoder-clustering architecture comprising: a feature encoder transforming surface area features into discriminative representations, a cluster assignment network generating probability distributions over target subregions, and an optimization framework jointly optimizing GCSD and symmetry losses. The framework processes bilateral datasets $X_L \in \mathbb{R}^{n \times m}$ and $X_R \in \mathbb{R}^{n \times m}$ containing original and symmetric features. Please refer to Figure 2 for the detailed architecture.

**Encoder.** The encoder of USDPnet employs a four-layer multilayer perceptron architecture consisting of an input layer of dimension $d_{\text{in}}$, two intermediate hidden layers each containing 500 units with ReLU activation functions, and an output layer of dimension $d_z$ with Tanh activation. Batch normalization is applied after each hidden layer. The original data $X$ is transformed by the feature encoder into distinguishable low-dimensional features $Z$. Specifically, the input data is organized into bilateral hemisphere datasets $X_L, X_R \in \mathbb{R}^{n \times m}$, where $n = n_L + n_R$ represents the total number of vertices across both hemispheres. Each dataset is constructed by concatenating original and symmetric features: $X_L = \{X_R^S, X_L^O\}$ and $X_R = \{X_R^O, X_L^S\}$, where the semicolon denotes vertical concatenation. The original features $X_L^O \in \mathbb{R}^{n_L \times m}$ and $X_R^O \in \mathbb{R}^{n_R \times m}$ represent local surface area features from left and right hemispheres, while symmetric features $X_L^S \in \mathbb{R}^{n_L \times m}$ and $X_R^S \in \mathbb{R}^{n_R \times m}$ are generated through symmetric mapping and one nearest neighbor matching. Notably, $X_R^S$ and $X_R^O$ maintain identical dimensions and occupy symmetric positions within $X_L$ and $X_R$, with $X_L^O$ and $X_L^S$ following the same dimensional and positional correspondence, as illustrated in Figure 1 (b) and (c). These extended datasets are fed into the same encoder to obtain corresponding low-dimensional features $Z_L \in \mathbb{R}^{n \times h}$ and $Z_R \in \mathbb{R}^{n \times h}$.

**Cluster Assignment Network.** The clustering module comprises a fully connected layer with 100 nodes, followed by Batchnorm layer,a ReLU activation and a Softmax layer. The network takes the low-dimensional feature embeddings $Z_L$ or $Z_R$ as inputs, and outputs the corresponding soft assignment matrices $A_L \in \mathbb{R}^{n \times r}$ and $A_R \in \mathbb{R}^{n \times r}$, where each row represents the probability distribution of a sample over all clusters. The composition of $A_L$ and $A_R$ corresponds to that of $X_L$ and $X_R$, comprising $A_L = \{A_R^S, A_L^O\}$ and $A_R = \{A_R^O, A_L^S\}$. Among these, $A_L^O \in \mathbb{R}^{n_L \times r}$ and $A_R^O \in \mathbb{R}^{n_R \times r}$ serve as the final assignment matrices applied to clustering results, where the features represent the ultimate vertex labels. The complete matrices $A_L$ and $A_R$ are utilized for subsequent loss function computation, as illustrated in Figure 2 (b).

For each subcortical nuclei, to achieve effective parcellation across multiple clusters, we adopt the maximization of GCSD among multiple clusters in the feature embeddings $Z_L$ or $Z_R$ as the principal clustering objective, combined with simplex regularization on the assignment matrix $A$:

$$\mathcal{L}_{\text{GCSD}} = D_{\text{GCS}}(Z, A) + \lambda_1 \operatorname{tr}(AA^\top) + \lambda_2 \operatorname{tr}(QQ^\top), \tag{1}$$

where $\lambda_1, \lambda_2 \geq 0$ are regularization coefficients balancing the trade-off between divergence maximization and cluster representation constraints, $A^\top$ and $Q^\top$ denote the matrix transpose of $A$ and $Q$, and $\operatorname{tr}(\cdot)$ represents the matrix trace operation. The core term $D_{\text{GCS}}$, representing the generalized Cauchy-Schwarz divergence, follows the estimator in Lu et al. (2025b):

$$D_{\text{GCS}}(Z, A) = -\log\left(\frac{1}{r}\operatorname{sum}\left(\frac{\left(A^{r-1}\right)^\top}{KA}\operatorname{prod}(KA)\right)\right) + \frac{1}{r}\operatorname{tr}\left[\log\left((A^\top)^{r-1}(KA)^{r-1}\right)\right], \tag{2}$$

where $K$ is the Gram matrix computed from the positive definite kernel $\kappa_\sigma$, such that $K_{ij} = \kappa_\sigma(\mathbf{z}_i - \mathbf{z}_j)$ for all $(i, j)$ pairs, and $r$ represents the dimensionality of the cluster space encoded by the assignment matrix $A$. The notation $\operatorname{sum}(\cdot)$ signifies the summation of all elements within a matrix, $\operatorname{prod}(A)$ calculates the product of row elements in matrix $A$, yielding a column vector. The second and third terms in equation 1 serve to promote orthogonality among the clusters and to enforce the proximity of the cluster membership vectors to a corner of the simplex. The proximity matrix $Q$ is derived from the assignment matrix $A$ as:

$$Q_{ij} = \exp\left(-\|\alpha_i - e_j\|^2\right), \tag{3}$$

where $\alpha_i$ denotes the $i$-th row of $A$ and $e_j$ denotes the $j$-th standard basis vector. Maximizing $\operatorname{tr}(QQ^\top)$ drives the assignment vectors toward one-hot form, thereby yielding more discriminative cluster assignments. Details on the regularization terms can be found in Kampffmeyer et al. (2019).

Compared to traditional pairwise divergence computation methods, GCSD offers significant advantages through joint entropy computation, enabling single joint computation across all distributions simultaneously, as illustrated in Figure 2 (b). This approach enhances clustering stability and computational efficiency by avoiding the need for multiple pairwise comparisons and providing a unified measure for multi-cluster optimization.

**Symmetry Constraint.** Motivated by the inherent anatomical symmetry of brain regions, we introduce a symmetry constraint to address common limitations in traditional methods: left-right parcellation inconsistency, poor stability, and high initialization sensitivity. Symmetry incorporation substantially improves result stability and reproducibility while reducing dependence on initial conditions. To leverage anatomical symmetry, we perform symmetric point matching between hemispheric data through midsagittal plane correspondence. Through symmetric coordinate mapping and nearest neighbor matching, we obtain augmented datasets $X_L^S$ and $X_R^S$ containing cross-hemisphere symmetric correspondences.The details have been mentioned above.

The encoder processes the concatenated datasets $X_L$ and $X_R$ to generate assignment matrices $A_L$ and $A_R$, where the *symmetry loss* employs Mean Squared Error (MSE) to quantify discrepancies between symmetric assignments:

$$\mathcal{L}_{\text{sym}} = MSE(A_L, A_R) = \frac{1}{n}\sum_{i=1}^{n}\|\alpha_{L,i} - \alpha_{R,i}\|^2, \tag{4}$$

where $\alpha_{L,i}$ and $\alpha_{R,i}$ denote the $i$-th rows of $A_L$ and $A_R$ (cluster probability vectors for corresponding samples), and $n$ is the total number of samples.

The overall loss function of the proposed framework is formulated as:

$$\mathcal{L} = \mathcal{L}_{\text{GCSD}}^{L} + \mathcal{L}_{\text{GCSD}}^{R} + \lambda_3 \, \mathcal{L}_{\text{sym}}, \tag{5}$$

where $\mathcal{L}_{\text{GCSD}}^{L}$ and $\mathcal{L}_{\text{GCSD}}^{R}$ denote the GCSD-based clustering losses for the left and right hemispheric feature embeddings, $\mathcal{L}_{\text{sym}}$ is the MSE-based symmetry constraint term, and $\lambda_3 > 0$ is a weighting coefficient controlling the strength of the symmetry regularization.

## 4 EXPERIMENTS

### 4.1 EXPERIMENTAL SETUP

**Dataset and Preprocessing.** We use MRI data from the Baby Connectome Project (BCP) (Howell et al., 2019) covering 513 typically developing infants aged 0-2 years. All preprocessing steps strictly followed the iBEAT V2.0 pipeline as described in (Wang et al., 2023), including intensity correction, skull stripping, registration, and surface reconstruction. Following the same methodology, we generated vertex-corresponding surface models for 6 subcortical nuclei (hippocampus, amygdala, thalamus, caudate, putamen, pallidum). Local surface areas at each vertex were computed across multiple developmental stages and Gaussian-smoothed to construct temporal feature vectors. This yields bilateral surface meshes where each vertex index encodes stage-specific area measurements, forming growth trajectory representations that capture morphological development patterns essential for age-sensitive parcellation, as illustrated in Figure 1 (a).

**Implementation Details.** The regularization weights were set to $\lambda_1 = \lambda_2 = 5 \times 10^{-2}$, while the symmetry term weight $\lambda_3 = 1 \times 10^{-1}$ was configured to prioritize symmetric convergence in clustering results. We employ 0.2 dropout probability for regularization. SGD optimizer is used with initial learning rate $1 \times 10^{-3}$ and momentum 0.9, coupled with ReduceLROnPlateau scheduler. The adaptive epoch strategy follows epochs $= r \times 1500$, where $r$ represents the target subregion count, with comprehensive experiments conducted across bilateral hemispheres for all 6 subcortical nuclei with subregion counts ranging from 2 to 10. To fully exploit GCSD's joint entropy computation capability, we adopt a full-batch training strategy using the complete dataset as a single batch, enabling global optimization over the entire data distribution and maximizing GCSD's theoretical advantages. Detailed implementation configurations are provided in the Appendix.

**Competing Methods and Metrics.** We compare our method against representative unsupervised clustering approaches across four categories: **(1) Matrix factorization methods** including Deep-NMF (Trigeorgis et al., 2014) and NMF (Wild et al., 2003); **(2) Spectral clustering variants** including Spectral + Discretization (Ng et al., 2001), Spectral + GMM (Azimbagirad & Junior, 2021), Spectral + K-Means (Sinaga & Yang, 2020), Spectral + K-Medoids (Park & Jun, 2009), and Spectral + QR (Narasimhan et al., 2005); **(3) Dimensionality reduction + clustering** including t-SNE + Agglomerative (Maaten & Hinton, 2008), UMAP + Agglomerative (McInnes et al., 2018), and PCA + Agglomerative (Abdi & Williams, 2010); **(4) Deep clustering** including DEC (Xie et al., 2016), DDC (Kampffmeyer et al., 2019), and GJRD (Lu et al., 2025a) which employs generalized Jensen-Rényi divergence to handle multiple distributions, providing a baseline for multi-distribution clustering comparison. We evaluate clustering performance using four metrics: Silhouette Coefficient (SC), Calinski-Harabasz Index (CH), Reconstruction Error (RE), and Feature Homogeneity (FH) (Rousseeuw, 1987; Caliński & Harabasz, 1974; Valle et al., 1999; Rosenberg & Hirschberg, 2007). As deep clustering methods are prone to local minima, a common problem for unsupervised deep architectures, we conduct 30 independent runs per configuration and report results with optimal unsupervised loss convergence. Baseline methods subject to random initialization variability undergo equivalent multiple runs ($\geq$30), with best-performing results selected for fair comparison.

### 4.2 RESULT ANALYSIS

**Clustering Performance Comparison.** Table 1 reports the best performance achieved by various methods under the optimal settings derived in Appendix D.4. These results demonstrate USDP-net's superiority across all metrics and both hemispheres. For cluster cohesion, USDPnet achieves the highest SC scores (0.328 left, 0.317 right), indicating tighter within-cluster similarity compared to the second-best Spectral + Discretization (0.311 left, 0.312 right) and significantly outperforming traditional methods like NMF (0.246 left, 0.244 right). For inter-cluster separability, USDPnet

Table 1: Performance comparison of unsupervised clustering methods and the ablation study of USDPnet. We use **bold** and underline text to denote the **first** and second places respectively.

| Method | Left Hemisphere | | | | Right Hemisphere | | | |
|---|---|---|---|---|---|---|---|---|
| | SC↑ | CH↑ | RE↓ | FH↑ | SC↑ | CH↑ | RE↓ | FH↑ |
| DeepNMF | 0.309 | 891 | 1.259 | 0.638 | 0.310 | 763 | 1.341 | 0.622 |
| NMF | 0.246 | 479 | 2.019 | 0.647 | 0.244 | 420 | 2.060 | 0.651 |
| Spectral+Disc | 0.311 | 989 | 1.187 | 0.668 | 0.312 | 919 | 1.216 | 0.661 |
| Spectral+GMM | 0.276 | 851 | 1.301 | 0.639 | 0.260 | 735 | 1.379 | 0.622 |
| Spectral+KMeans | 0.296 | 985 | 1.192 | 0.677 | 0.299 | 922 | 1.216 | 0.673 |
| Spectral+KMed | 0.307 | 900 | 1.253 | 0.665 | 0.308 | 791 | 1.325 | 0.651 |
| Spectral+QR | 0.280 | 961 | 1.210 | 0.666 | 0.278 | 892 | 1.242 | 0.657 |
| tSNE+Agglo | 0.219 | 766 | 1.368 | 0.648 | 0.252 | 725 | 1.365 | 0.629 |
| UMAP+Agglo | 0.271 | 833 | 1.307 | 0.654 | 0.299 | 728 | 1.371 | 0.633 |
| PCA+Agglo | 0.111 | 1142 | 1.096 | 0.701 | 0.107 | 1088 | 1.096 | 0.700 |
| DEC | 0.247 | 1055 | 1.145 | 0.681 | 0.159 | 953 | 1.196 | 0.676 |
| DDC | 0.243 | 1195 | 1.092 | 0.702 | 0.255 | 1003 | 1.167 | 0.678 |
| GJRD | 0.252 | 1118 | 1.100 | 0.697 | 0.178 | 976 | 1.174 | 0.686 |
| **Ours** w/o $A$ regularization | 0.312 | 1189 | 1.085 | 0.667 | 0.309 | 1091 | 1.092 | 0.665 |
| **Ours** w/o $Q$ regularization | 0.320 | 1113 | 1.105 | 0.669 | 0.311 | 1049 | 1.112 | 0.672 |
| **Ours** w/o $\mathcal{L}_{\text{sym}}$ | 0.306 | 1132 | 1.115 | 0.688 | 0.305 | 1049 | 1.121 | 0.681 |
| **Ours** | **0.328** | **1228** | **1.083** | **0.708** | **0.317** | **1123** | **1.087** | **0.704** |

attains the highest CH values (1228 left, 1123 right), demonstrating superior cluster distinctiveness compared to DDC (1195 left, 1003 right) and other deep methods. For reconstruction fidelity, USDPnet achieves the lowest RE (1.083 left, 1.087 right), indicating minimal information loss during feature compression. For anatomical consistency, USDPnet maintains the highest FH (0.708 left, 0.704 right), ensuring parcellated subregions preserve meaningful neuroanatomical boundaries. These findings highlight USDPnet's capacity to jointly optimize multiple dimensions of parcellation quality, demonstrating the effectiveness of GCSD-based clustering and symmetry constraints in achieving anatomically coherent subcortical delineations.

To further validate the robustness and stability of USDPnet, we present average performance results with standard deviation statistics in Table 2. These results are averaged across bilateral hemispheres and subregion counts 2-10 over 30 independent runs for each method, where all baseline methods are subject to random initialization effects. The analysis shows that USDPnet consistently outperforms all baseline methods across all evaluation metrics, demonstrating superior clustering quality with lower variability compared to traditional methods.

**Statistical Significance Analysis.** To further substantiate the performance advantages, we conducted Welch's t-tests comparing USDPnet against leading deep clustering methods (DEC, DDC, GJRD) under the same 30 experimental runs. The analysis reveals statistically significant superiority on all metrics ($p < 0.05$ for all 12 comparisons), with large effect sizes (Cohen's $d > 0.8$) in 11/12 comparisons demonstrating both statistical significance and practical importance. USDPnet shows clear advantages over DEC ($d = 2.530$ for SC, $d = 1.748$ for FH), substantial improvements over DDC ($d = 1.248$ for SC, $d = 1.474$ for FH), and significant gains over GJRD ($d = 1.142$ for SC, $d = 1.254$ for FH). After Bonferroni correction ($\alpha = 0.0042$), 91.7% of comparisons remain significant, providing robust evidence of USDPnet's superiority over SOTA approaches.

**Parcellation Visualization Analysis.** Through systematic analysis of clustering metric variations across different target subregion numbers for each nucleus, particularly examining SC coefficient trends (detailed in Appendix), we determined optimal clustering numbers for each nucleus: Hippocampus (7), Amygdala (2), Thalamus (10), Caudate (5), Putamen (4), and Pallidum (2). All comparative methods were evaluated under these metric-driven configurations. Figure 3 presents a comprehensive qualitative comparison of USDPnet against representative baseline methods: Spectral + Discretization (top SC performer), DeepNMF (established matrix factorization approach), DDC (recent deep clustering method), and Tian et al. (2020)'s functional connectivity gradient-based parcellation using voxel data. The visualization reveals critical limitations in existing approaches that USDPnet effectively addresses. In the baseline methods, several hemispheric discrepancies and irregular parcellation fragments are observed (marked with red circles), indicating inconsistent clustering behavior across methods.

Table 2: Average performance comparison across clustering methods with standard deviations (averaged across bilateral hemisphere regions for subregion counts 2-10 over 30 independent runs).We use **bold** text to denote the **first** places.

| Method | SC↑ | CH↑ | RE↓ | FH↑ |
|---|---|---|---|---|
| Spectral+GMM | 0.227±0.058 | 644±202 | 1.585±0.431 | 0.578±0.090 |
| Spectral+KMeans | 0.239±0.059 | 747±267 | 1.423±0.269 | 0.614±0.069 |
| Spectral+KMed | 0.234±0.068 | 563±278 | 1.688±0.446 | 0.563±0.106 |
| tSNE+Agglo | 0.208±0.024 | 659±90 | 1.472±0.128 | 0.608±0.029 |
| UMAP+Agglo | 0.220±0.038 | 631±100 | 1.507±0.130 | 0.594±0.035 |
| DEC | 0.135±0.052 | 866±139 | 1.304±0.160 | 0.643±0.040 |
| DDC | 0.187±0.074 | 885±149 | 1.289±0.148 | 0.648±0.040 |
| GJRD | 0.203±0.049 | 952±134 | 1.224±0.117 | 0.676±0.034 |
| **Ours** | **0.253±0.065** | **1035±127** | **1.161±0.084** | **0.712±0.041** |

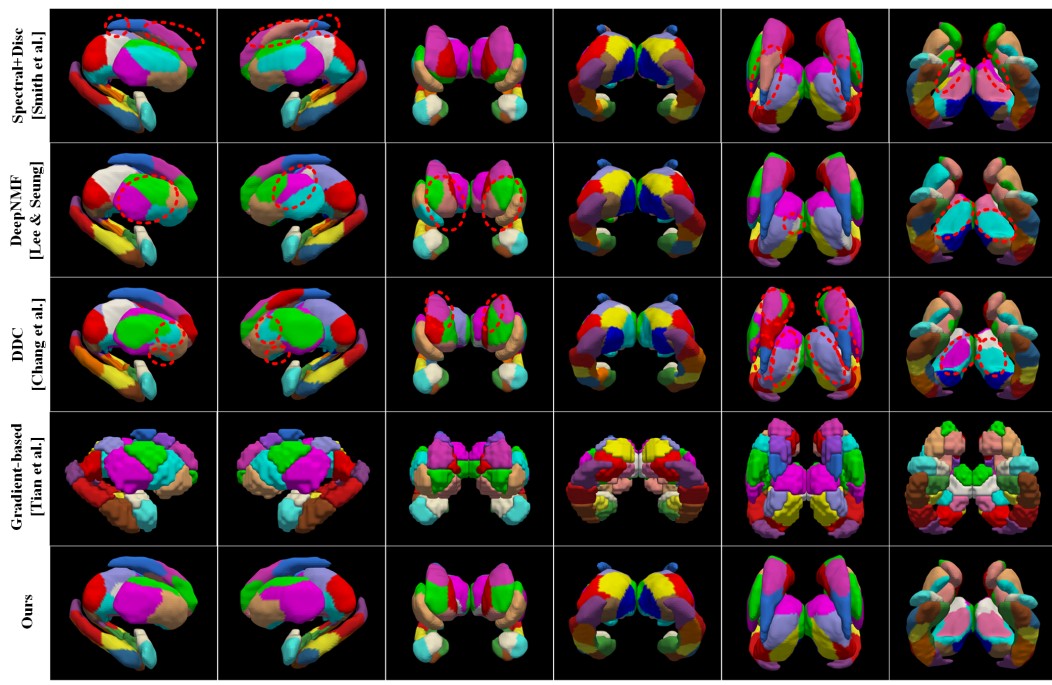

Figure 3: Qualitative comparison of subcortical parcellation results between USDPnet and representative baseline methods. Visualizations show bilateral parcellations across multiple nuclei. Compared to baseline methods and functional connectivity gradient parcellation (Tian et al., 2020)-USDPnet yields anatomically coherent, symmetric subregions without spurious outliers or inter-hemispheric inconsistencies (highlighted with red circles). The resulting parcellations exhibit strong alignment with expert-defined anatomical boundaries and SOTA adult parcellation (Tian et al., 2020), demonstrating both anatomical consistency and developmental relevance.

These anomalous results reflect the limitations of existing methods, including their susceptibility to initialization variance, inability to enforce anatomical constraints, and limited capacity to extract meaningful developmental features from complex infant brain data. In contrast, USDPnet produces more bilaterally consistent and symmetric parcellation boundaries than baseline methods, and its surface area-based parcellations visually align with the voxel-level gradients reported in Tian et al. (2020), indicating agreement in the captured spatial patterns. This hemispheric consistency and pattern-level concordance supports USDPnet as a robust clustering solution that effectively exploits surface-based feature representations, which are more cost-effective and scalable to acquire from standard sMRI than modalities such as fMRI or individualized anatomical delineations, particularly in large infant cohorts.

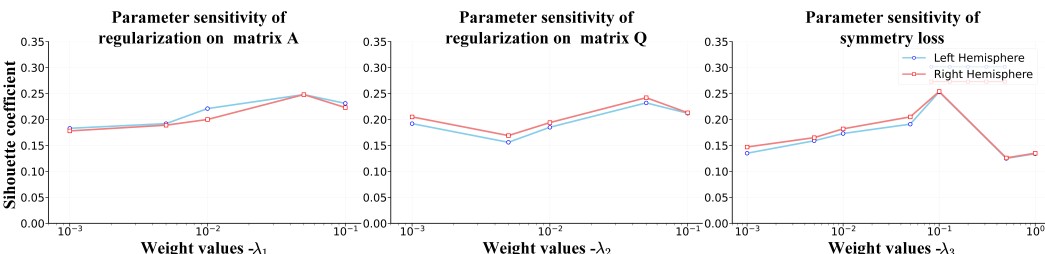

Figure 4: Parameter sensitivity analysis for regularization weights $\lambda_1$, $\lambda_2$, and $\lambda_3$. The vertical axis shows the silhouette coefficient (SC) and the horizontal axis shows the corresponding weight values.

Table 3: Comparison results of different divergences. We use **bold** text to denote the **first** places.

| Method | Left Hemisphere | | | | Right Hemisphere | | | |
|---|---|---|---|---|---|---|---|---|
| | SC ↑ | CH ↑ | RE ↓ | FH ↑ | SC ↑ | CH ↑ | RE ↓ | FH ↑ |
| CSD | 0.252 | 1075 | 1.149 | 0.689 | 0.255 | 1003 | 1.167 | 0.678 |
| KLD | 0.218 | 1056 | 1.149 | 0.680 | 0.200 | 1007 | 1.145 | 0.676 |
| GJRD-1(JSD) | 0.281 | 965 | 1.212 | 0.664 | 0.274 | 865 | 1.255 | 0.653 |
| GJRD-2 | 0.264 | 970 | 1.208 | 0.668 | 0.265 | 895 | 1.228 | 0.656 |
| **Ours**(GCSD) | **0.328** | **1228** | **1.042** | **0.708** | **0.317** | **1123** | **1.087** | **0.704** |

Overall, our approach establishes a robust and interpretable foundation for analyzing early infant brain development. By providing developmentally grounded, anatomically symmetric, and functionally relevant subregional delineations, our subcortical parcellations not only visualizes the fine-grained organization of subcortical structures during the first two years of life—a period characterized by heightened neurodevelopmental plasticity and critical windows for the maturation of cognitive, emotional, and sensorimotor functions—but also supports research into how early subcortical architecture scaffolds the formation of neural circuits and guides subsequent behavioral development, providing a robust foundation for mechanistic investigations into the emergence and differentiation of brain functions. Beyond its neuroscientific implications, USDPnet opens promising avenues for early identification and personalized intervention of neurodevelopmental disorders. Fine-scale, symmetric subcortical parcellations can guide the discovery of atypical growth patterns associated with conditions such as ASD and ADHD. Moreover, our anatomically informed parcellation results provide a structural foundation for refining precision DBS targets, enabling more individualized and developmentally appropriate neuromodulation therapies.

**Ablation Study.** We conducted comprehensive ablation analysis to evaluate the contribution of each loss component by systematically removing: (1) orthogonality regularization on Matrix $A$ (setting $\lambda_1 = 0$), (2) simplex regularization on row vectors of matrix $A$ through $Q$ (setting $\lambda_2 = 0$), and (3) symmetry loss term (setting $\lambda_3 = 0$). Table 1 presents the results across all clustering metrics for bilateral hemispheres. The analysis reveals that removing the symmetry constraint yields the most substantial performance degradation across all metrics, confirming its critical role in maintaining anatomical consistency. Matrix regularization terms show moderate but consistent impacts, with $Q$ regularization contributing more significantly to clustering stability than $A$ regularization.

**Favorable Parameter Settings.** We identified favorable parameter settings through systematic analysis on the thalamus using 30 seeds across logarithmically-spaced values. Results indicate that clustering performance (silhouette coefficient) remains stable within a specific range but degrades when weights deviate significantly. A favorable configuration is $\lambda_1 = \lambda_2 = 5 \times 10^{-2}$ and $\lambda_3 = 1 \times 10^{-1}$ (Figure 4). This setting maintains balanced weights, avoiding excessive symmetry loss that would overemphasize bilateral matching and yield anatomically implausible parcellations. Additional details are available in the Appendix B.3.

**Comparison of different divergences.** We evaluated Cauchy-Schw_arz (CSD), Kullback-Leibler (KLD), and generalized Jensen-Rényi (GJRD) divergences, with quantitative results in Table 3. Unlike the GJRD baseline in Table 1, these variants modify only the divergence term of USDPnet while

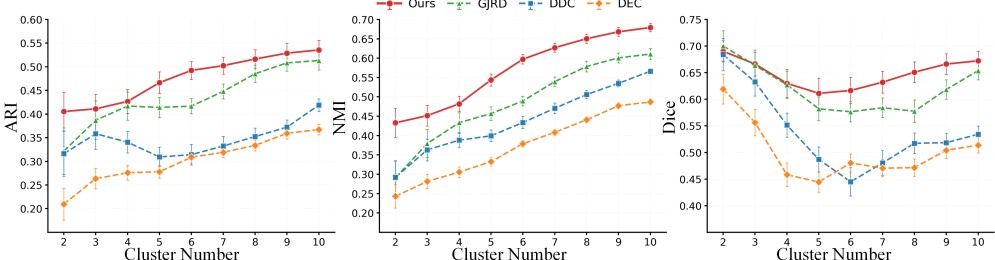

Figure 5: Reproducibility analysis results for various clustering methods. The vertical axis represents clustering metrics (Dice, NMI, ARI) and the horizontal axis shows different methods. Error bars indicate standard deviation across 30 runs.

Table 4: Comparison of symmetry-preserving strategies for subcortical parcellation. We use **bold** to denote the best results.

| Symmetrization Strategy | Left Hemisphere | | | | Right Hemisphere | | | |
|---|---|---|---|---|---|---|---|---|
| | SC ↑ | CH ↑ | RE ↓ | FH ↑ | SC ↑ | CH ↑ | RE ↓ | FH ↑ |
| Input-level | 0.303 | 1129 | 1.116 | 0.686 | 0.302 | **1130** | 1.116 | 0.688 |
| Output-level | 0.288 | 1119 | 1.212 | 0.663 | 0.289 | 1119 | 1.211 | 0.664 |
| **Ours (In-training)** | **0.328** | **1228** | **1.042** | **0.708** | **0.317** | 1123 | **1.087** | **0.704** |

keeping the architecture fixed. GJRD-1 and GJRD-2 correspond to Rényi orders 1 and 2, with order 1 reducing to Jensen-Shannon divergence (JSD). Across all configurations, GCSD achieves superior clustering quality and computational efficiency, which may contribute to robustness in infant brain imaging applications.

**Reproducibility Analysis.** We evaluated reproducibility for deep clustering methods across 30 runs on thalamus with $r \in \{2, \ldots, 10\}$ subregions (Figure 5). We assessed weighted Dice for spatial overlap, NMI for normalized mutual information, and ARI for chance-corrected pairwise agreement. Most methods show good reproducibility at 2 regions, but it drops sharply at 3-4 regions and then gradually declines. In contrast, our method consistently yields superior performance and exhibits relative stability at higher region numbers.

**Comparison of Symmetry-Preserving Strategies.** We compared our in-training soft symmetry regularization against two alternatives: input-level symmetrization (averaging and mirroring data before training) and output-level symmetrization (averaging assignment matrices after independent training). Results in Table 4 show that harder constraints degrade performance, confirming that our method discovers latent bilateral structure rather than inflating metrics through enforced homogeneity. Additional details are provided in Appendix D.6.

## 5 CONCLUSION

We propose USDPnet, an Unsupervised Symmetric Deep Parcellation Network for infant subcortical nuclei delineation in early postnatal development. By combining deep autoencoder-based embedding with a clustering objective driven by generalized Cauchy–Schwarz divergence (GCSD), USDPnet enhances inter-cluster separability in high-dimensional developmental features. A symmetry regularization term ensures bilateral consistency and mitigates initialization bias. Experiments show that USDPnet outperforms classical and state-of-the-art clustering methods in both accuracy and anatomical coherence. Visualizations reveal strong alignment with known anatomical boundaries and functional connectivity, underscoring anatomical consistency. While surface area serves as a discriminative morphological feature, its limitations highlight the need for multimodal integration and denser temporal sampling in future work. **Overall** USDPnet offers a robust, anatomically informed, and computationally efficient framework for infant subcortical parcellation, with broad applicability to other symmetric brain regions and early neurodevelopmental research.

## 6 ETHICS AND REPRODUCIBILITY STATEMENT

This study used infant MRI data from the Baby Connectome Project (BCP), openly available, with parental consent and ethical approval. The proposed parcellation approach is intended only for research purposes and requires expert validation for clinical applicability. We commit to open-sourcing the code to ensure reproducibility and responsible use. **Declaration of using LLMs**: LLMs were only used for language editing. All scientific content, analysis, and results are originally produced.

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

## A  DECLARATION OF LARGE LANGUAGE MODELS (LLMS) USAGE

We only employed LLMs to enhance manuscript quality through grammar correction, error iden-
tification, and clarity optimization. All AI-generated suggestions underwent rigorous human re-
view and adaptation. The authors retain full responsibility for all content and conclusions presented
herein.

## B  REPRODUCIBILITY

### B.1  DETAILED IMPLEMENTATION CONFIGURATION

**Computational Environment.** All experiments were conducted on Ubuntu 20.04 with Python 3.8.8
and PyTorch framework, accelerated using CUDA 12.2 for efficient GPU computation.

**Hyperparameters.** We configured regularization weights $\lambda_1 = \lambda_2 = 5 \times 10^{-2}$ and symmetry
weight $\lambda_3 = 1 \times 10^{-1}$ to prioritize anatomically-consistent bilateral convergence. The model em-
ployed $p = 0.2$ dropout probability and SGD optimizer with learning rate $\alpha_0 = 1 \times 10^{-3}$, momen-
tum $\mu = 0.9$. ReduceLROnPlateau scheduler utilized reduction factor $\gamma = 0.5$, patience $P = 1000$
epochs, cooldown $C = 50$ epochs, and minimum learning rate $\alpha_{min} = 1 \times 10^{-3}$.

**Training Protocol.** The adaptive epoch strategy followed epochs $= r \times 1500$ where $r$ represents
target subregion count. Full-batch training was implemented to maximize GCSD's joint entropy
computation advantages, followed by $n_{finetune} = 30$ fine-tuning iterations for optimal performance
across all six subcortical nuclei (amygdala, caudate nucleus, hippocampus, pallidum, putamen, and
thalamus) with bilateral hemispheric training for both left and right structures, encompassing subre-
gional configurations $r \in \{2, 3, ..., 10\}$ for comprehensive parcellation analysis.

### B.2  BASELINE METHODS CONFIGURATION

**Fair Comparison Setup.** All baseline methods were evaluated using identical data preprocessing
protocols, and evaluation metrics to ensure fair comparison. Each method underwent 30 independent
runs with different random initializations to account for stochastic variability.

**Deep Learning Baselines.** DEC (Xie et al., 2016), DDC (Kampffmeyer et al., 2019), and GJRD
(Lu et al., 2025a) were implemented using their original architectures with learning rates adapted to
$1 \times 10^{-3}$ for consistent convergence. GJRD employed identical encoder architecture to our method
for fair feature comparison.

**Traditional Methods.** All methods utilized L2-normalized input features. *Agglomerative Clus-
tering* employed Ward linkage with Euclidean distance. *NMF variants* used nndsvd initialization,
maximum iterations = 80000, tolerance = $1 \times 10^{-4}$, regularization coefficient $\alpha = 0.1$, and L1-
ratio = 0.5. *DeepNMF* implemented hierarchical matrix factorization with random initialization,
maximum iterations = 80000, tolerance = $1 \times 10^{-5}$, Frobenius beta loss, regularization coeffi-
cients $\alpha_W = \alpha_H = 0.01$, L1-ratio = 0.0, and Min-Max normalization preprocessing followed by
K-means clustering on latent representations. *Spectral Clustering* variants configured RBF kernel
with $\gamma = 1.0$ and discretize assignment strategy. Dimensionality reduction methods (PCA, t-SNE,
UMAP) used default scikit-learn parameters followed by Ward agglomerative clustering.

### B.3  FAVORABLE PARAMETER SETTINGS DETAILS

To determine the favorable configuration of regularization weights, we conducted a comprehensive
parameter analysis on the thalamus with 10 target clusters. We initialized the baseline configura-
tion with $\lambda_1 = \lambda_2 = \lambda_3 = 1 \times 10^{-2}$ and the weight coefficient for $D_{GCS}$ fixed at unity. For
this assessment, we defined 12 logarithmically-spaced values spanning five orders of magnitude:
$\{1 \times 10^{-5}, 5 \times 10^{-5}, 1 \times 10^{-4}, 5 \times 10^{-4}, 1 \times 10^{-3}, 5 \times 10^{-3}, 1 \times 10^{-2}, 5 \times 10^{-2}, 1 \times 10^{-1}, 5 \times 10^{-1}, 1, 5\}$. For each parameter $\lambda_i$ ($i \in \{1, 2, 3\}$), we systematically varied its value across the
12-point grid while maintaining the remaining two parameters at the baseline value of $1 \times 10^{-2}$.
To ensure statistical reliability, we employed 30 high-performing initialization seeds consistently

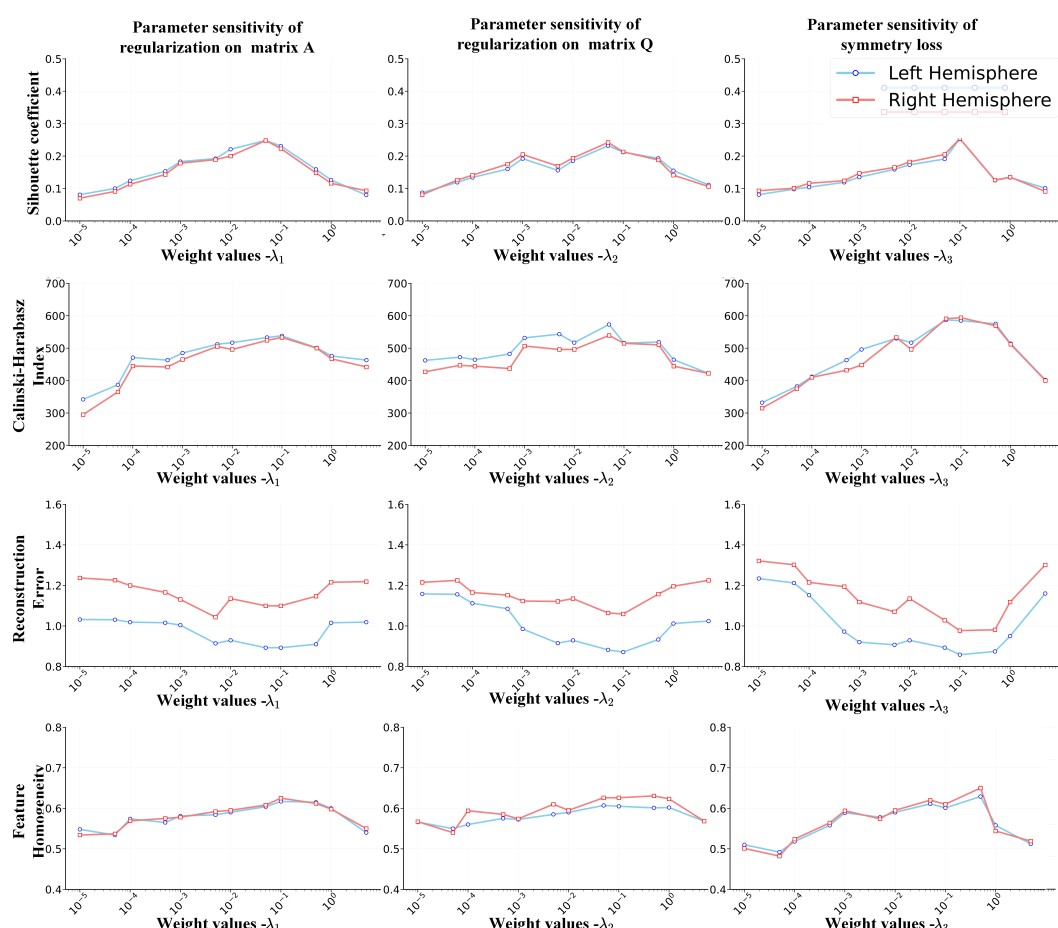

Figure 6: Parameter analysis for regularization weights $\lambda_1$, $\lambda_2$, and $\lambda_3$. Each plot illustrates the effect of varying one parameter across a logarithmic scale on clustering performance, evaluated using four metrics: SC, RE, CH, and FH. Performance scores are averaged over 30 independent runs. This analysis identifies the favorable parameter configuration that ensures stable and high-quality clustering.

across all parameter configurations. For each experimental condition, we executed clustering procedures and computed SC, RE, CH, and FH metrics for both hemispheres. Performance evaluation was based on the mean values of these metrics averaged across 30 independent runs and bilateral hemispheres on the thalamus. The performance profiles for all three parameters are visualized in Figure 6, enabling the identification of favorable parameter ranges. Based on these empirical results, we identified a favorable configuration as $\lambda_1 = \lambda_2 = 5 \times 10^{-2}$ and $\lambda_3 = 1 \times 10^{-1}$, which demonstrated superior clustering performance for subsequent experiments. All other experimental settings remained consistent with those specified in the Experimental Setup 4.1 section.

## B.4 COMPUTATIONAL RESOURCE CONSUMPTION

This section summarizes the computational resources and runtime required for model training and experiments. All experiments were performed on a system equipped with NVIDIA RTX 3090 GPUs. Table 5 reports the number of vertices, which represents the data size $n$ in $\{x_i \in X\}_{i=1}^{n}$, for each nucleus in the left and right hemispheres. Each vertex is associated with a 513-dimensional feature vector, corresponding to the dimension of $x_i \in \mathbb{R}^m$, where each dimension encodes the surface area of the vertex at a specific age in the atlas. We further evaluated our method on all six nuclei

Table 5: Vertex counts, running time (epochs/s), and peak GPU memory usage (GB) for subcortical nuclei in left and right hemispheres.

| Metric | Subcortical Nuclei | | | | | |
|---|---|---|---|---|---|---|
| | Amyg. | Caud. | Hipp. | Pall. | Put. | Thal. |
| Vertices (Left) | 674 | 2,230 | 2,264 | 1,088 | 2,230 | 3,416 |
| Vertices (Right) | 694 | 2,176 | 2,332 | 1,100 | 2,234 | 3,346 |
| Running time (epochs/s) | 50.6 | 27.2 | 24.3 | 26.9 | 49.3 | 14.1 |
| Memory usage (GB) | 0.13 | 0.83 | 0.90 | 0.25 | 0.83 | 1.80 |

at their respective optimal cluster numbers and recorded the corresponding computational costs, as detailed in the table.

## C   EVALUATION METRICS FORMULATION

**Silhouette Coefficient (SC).** We employ a customized affinity-based silhouette coefficient formulation (Rousseeuw, 1987) to accommodate the inherent structure of subcortical surface area features. For each vertex $i$, the silhouette score $s_i$ is computed as:

$$s_i = \frac{b_i - a_i}{\max(a_i, b_i)} \tag{6}$$

where $a_i$ denotes the intra-cluster average dissimilarity: $a_i = \frac{1}{|C_i|-1} \sum_{j \in C_i, j \neq i} d(i,j)$, and $b_i$ represents the minimum inter-cluster average dissimilarity: $b_i = \min_{k \neq \text{cluster}(i)} \frac{1}{|C_k|} \sum_{j \in C_k} d(i,j)$. The dissimilarity metric is derived from the original feature affinity matrix as $d(i,j) = 1 - \mathcal{R}_{ij}$, where $\mathcal{R}$ represents the affinity matrix constructed from raw vertex area features. The overall silhouette coefficient is obtained by averaging across all vertices: $SC = \frac{1}{n} \sum_{i=1}^{n} s_i$.

**Calinski-Harabasz Index (CH).** This index (Caliński & Harabasz, 1974) quantifies the ratio of between-cluster to within-cluster variance, computed using the standard formulation:

$$CH = \frac{\text{tr}(\mathbf{B}_k)/(K-1)}{\text{tr}(\mathbf{W}_k)/(n-K)} \tag{7}$$

where $\mathbf{B}_k = \sum_{i=1}^{K} n_i (\boldsymbol{\mu}_i - \boldsymbol{\mu})(\boldsymbol{\mu}_i - \boldsymbol{\mu})^T$ constitutes the between-cluster scatter matrix, $\mathbf{W}_k = \sum_{i=1}^{K} \sum_{\mathbf{x} \in C_i} (\mathbf{x} - \boldsymbol{\mu}_i)(\mathbf{x} - \boldsymbol{\mu}_i)^T$ defines the within-cluster scatter matrix, $\boldsymbol{\mu}$ denotes the global feature centroid, and $\boldsymbol{\mu}_i$ represents the centroid of the $i$-th cluster.

**Reconstruction Error (RE).** This metric (Valle et al., 1999) evaluates clustering fidelity by quantifying the mean squared deviation between vertex features and their respective cluster centroids:

$$RE = \frac{1}{n} \sum_{k=1}^{K} \sum_{i \in C_k} ||\mathbf{x}_i - \boldsymbol{\mu}_k||^2 \tag{8}$$

where $\boldsymbol{\mu}_k = \frac{1}{|C_k|} \sum_{i \in C_k} \mathbf{x}_i$ denotes the centroid of the $k$-th cluster, computed as the arithmetic mean of all vertices assigned to that cluster.

**Feature Homogeneity (FH).** This coefficient (Rosenberg & Hirschberg, 2007) assesses intra-cluster feature consistency relative to global feature variability, formulated as:

$$FH = 1 - \frac{\bar{\sigma}_{\text{intra}}^2}{\sigma_{\text{global}}^2} \tag{9}$$

where $\bar{\sigma}_{\text{intra}}^2 = \frac{1}{K} \sum_{k=1}^{K} \frac{1}{d} \sum_{j=1}^{d} \text{Var}(\mathbf{X}_{C_k, j})$ represents the average intra-cluster variance across all feature dimensions, and $\sigma_{\text{global}}^2 = \frac{1}{d} \sum_{j=1}^{d} \text{Var}(\mathbf{X}_j)$ characterizes the global feature variance. Higher FH values indicate superior feature consistency within clusters relative to the overall dataset distribution.

Table 6: Comparison of symmetry weight scheduling strategies across left and right hemispheres. We use **bold** text to denote the **first** places.

| Scheduling strategy | Left Hemisphere | | | | Right Hemisphere | | | |
|---|---|---|---|---|---|---|---|---|
| | SC ↑ | CH ↑ | RE ↓ | FH ↑ | SC ↑ | CH ↑ | RE ↓ | FH ↑ |
| Cosine | 0.324 | 1211 | **1.038** | 0.696 | **0.320** | 1120 | 1.089 | 0.702 |
| Linear | 0.315 | 1154 | 1.102 | 0.692 | 0.311 | **1146** | 1.110 | 0.688 |
| Exponential | 0.311 | 1142 | 1.112 | 0.686 | 0.309 | 1055 | 1.119 | 0.679 |
| **Ours** | **0.328** | **1228** | 1.042 | **0.708** | 0.317 | 1123 | **1.087** | **0.704** |

# D SUPPLEMENTARY EXPERIMENTS

## D.1 SCHEDULING OF SYMMETRY LOSS WEIGHT.

Given that subtle asymmetries persist between the left and right hemispheres of subcortical nuclei, we designed a temporal scheduling strategy to dynamically adjust the symmetry constraint during training. We kept the random seed and all experimental settings identical to those used in Table 1, modifying only the symmetry weight scheduling strategies. A dedicated scheduler scaled the symmetry loss term, and we considered three schemes: cosine, linear, and exponential.

In all runs, the symmetry weight $w_{\mathrm{sym}}(t)$ was updated from an initial value $w_0$ (e.g., $w_0 = 2 \times 10^{-3}$) toward a final value $w_f$ (e.g., $w_f = 0$) over $T$ epochs, with epoch index $t$ and normalized progress $p = t/T$. The scheduler was active only for $t \geq t_{\mathrm{start}}$; otherwise $w_{\mathrm{sym}}(t) = w_0$. For the linear schedule, we used

$$w_{\mathrm{sym}}(t) = (1 - p)\, w_0 + p\, w_f, \tag{10}$$

for the cosine schedule

$$w_{\mathrm{sym}}(t) = w_f + (w_0 - w_f)\, \frac{1}{2}\big(1 + \cos(\pi p)\big), \tag{11}$$

and for the exponential schedule

$$w_{\mathrm{sym}}(t) = w_0 \exp(-\lambda p), \qquad \lambda = \begin{cases} -\ln\left(\dfrac{w_f}{w_0}\right), & w_f > 0, \\ \lambda_0, & w_f = 0, \end{cases} \tag{12}$$

where $\lambda_0$ is a fixed decay rate (in our experiments chosen such that the weight rapidly approaches zero near the end of training).

This setup aims to impose stronger bilateral consistency early while gradually relaxing the constraint to permit mild asymmetries. However, the results in Table 6 show that reducing the symmetry loss weight over training degrades performance and does not improve clustering quality. In practice, an appropriately chosen fixed symmetry weight better preserves global bilateral symmetry while still accommodating localized, neurobiologically meaningful asymmetries.

## D.2 VALIDATION OF THE FULL-BATCH TRAINING STRATEGY

We evaluated the effectiveness of the full-batch training strategy by systematically comparing it with stochastic gradient descent (SGD) using fixed mini-batch sizes of 64, 256, and 1,024. We kept the random seed and all experimental settings identical to those in Table 1, varying only the batch strategy, and conducted the experiments exclusively on the thalamus with the subregion number fixed at 10 (r = 10). The resulting clustering metrics are summarized in Table 7. Notably, the thalamus comprises 3416 vertices in the left hemisphere and 3346 vertices in the right hemisphere.

Table 7: Comparison results of different batch strategy across left and right thalamus(r=10) hemispheres.We use **bold** text to denote the **first** places.

| Batch strategy | Left Hemisphere | | | | Right Hemisphere | | | |
|---|---|---|---|---|---|---|---|---|
| | SC ↑ | CH ↑ | RE ↓ | FH ↑ | SC ↑ | CH ↑ | RE ↓ | FH ↑ |
| Batchsize=64 | 0.085 | 564 | 0.859 | **0.675** | 0.101 | 620 | 0.849 | **0.703** |
| Batchsize=256 | 0.223 | **647** | 0.791 | 0.647 | 0.182 | **708** | 0.857 | 0.687 |
| Batchsize=1024 | 0.281 | 645 | 0.792 | 0.644 | 0.231 | 661 | 0.897 | 0.645 |
| **Ours(Full batch)** | **0.285** | 644 | **0.792** | 0.631 | **0.294** | 648 | **0.908** | 0.642 |

Under the full-batch setting, each training pass processes all vertices simultaneously, yielding input matrices of size $X_L, X_R \in \mathbb{R}^{6762 \times m}$. In contrast, when employing fixed mini-batch strategies, each iteration uses batches of size $2 \times$batchsize for the concatenated bilateral inputs $X_L$ and $X_R$. The full-batch approach provides more stable gradient estimates and more faithfully captures the global data distribution, which is essential for optimizing the GCSD-based objective. Conversely, mini-batch SGD introduces stochastic noise due to sampling variability, which can hinder convergence and reduce consistency across runs. Empirically, our results demonstrate that full-batch training yields more reliable and reproducible clustering performance, particularly for the structurally complex thalamus, where subtle anatomical variations can substantially influence parcellation outcomes.

### D.3 COMPARISON OF SYMMETRY LOSS BASED ON MSE AND CROSS-ENTROPY

To compare the effectiveness of symmetry loss computed using MSE versus cross-entropy, we kept the random seed and all experimental settings identical to those used in Table 1, modifying only the formulation of the symmetry loss to cross-entropy. The results, summarized in Table 8, show that the MSE-based symmetry loss consistently outperforms its cross-entropy counterpart. A plausible explanation is that MSE provides a smoother and more stable gradient signal for enforcing bilateral correspondence, whereas cross-entropy tends to be more sensitive to local prediction fluctuations, leading to less reliable symmetry constraints.

### D.4 DETERMINATION OF OPTIMAL SUBREGION NUMBERS

To systematically determine the optimal parcellation granularity for each subcortical nucleus, we conducted comprehensive cluster validation analysis across subregion numbers ranging from $r = 2$ to $r = 10$. We evaluated the results from the 30 independent runs previously performed for all six subcortical nuclei across bilateral hemispheres, following the experimental protocol described in the Section 4.1, to ensure statistical robustness. The silhouette coefficient (SC) was computed for all clustering results, with average SC values calculated across all nuclei at each cluster number and visualized as line plots in Figure 7. The SC serves as our primary validation metric, quantifying clustering quality through the ratio of inter-cluster separation to intra-cluster compactness, where higher values indicate superior parcellation coherence and distinctiveness. Figure 7 establishes the criterion for selecting optimal subregion numbers, with arrows indicating regions of locally maximal SC values corresponding to the most appropriate parcellation granularities.

Our analysis revealed distinct SC optimization profiles reflecting the inherent structural complexity of each subcortical nucleus. The amygdala and pallidum demonstrated optimal SC values at $r = 2$, consistent with their relatively lower structural complexity and smaller volumetric characteristics, making them unsuitable for finer-grained subdivisions. Conversely, the caudate nucleus, hippocampus, and putamen achieved optimal SC values at $r = 5$, $r = 7$, and $r = 4$, respectively, reflecting their intermediate organizational complexity. The thalamus exhibited optimal SC performance at $r = 10$, reflecting its highly complex internal architecture. To validate our methodology's performance at higher granularities and definitively establish the optimal thalamic parcellation, we extended the analysis to encompass $r \in \{11, 12, \ldots, 15\}$. This extended evaluation confirmed that $r = 10$ remained optimal for thalamic parcellation, demonstrating the stability of our optimization approach. Based on this comprehensive validation analysis, we established the following optimal

Table 8: Comparison of MSE-based and cross-entropy–based symmetry loss formulations across left and right thalamus(r=10) hemispheres

| Batch strategy | Left Hemisphere | | | | Right Hemisphere | | | |
|---|---|---|---|---|---|---|---|---|
| | SC ↑ | CH ↑ | RE ↓ | FH ↑ | SC ↑ | CH ↑ | RE ↓ | FH ↑ |
| Cross-entry | 0.292 | 969 | 1.135 | 0.709 | 0.300 | 1030 | 1.146 | 0.703 |
| **Ours**(MSE) | **0.328** | **1228** | **1.042** | **0.708** | **0.317** | **1123** | **1.087** | **0.704** |

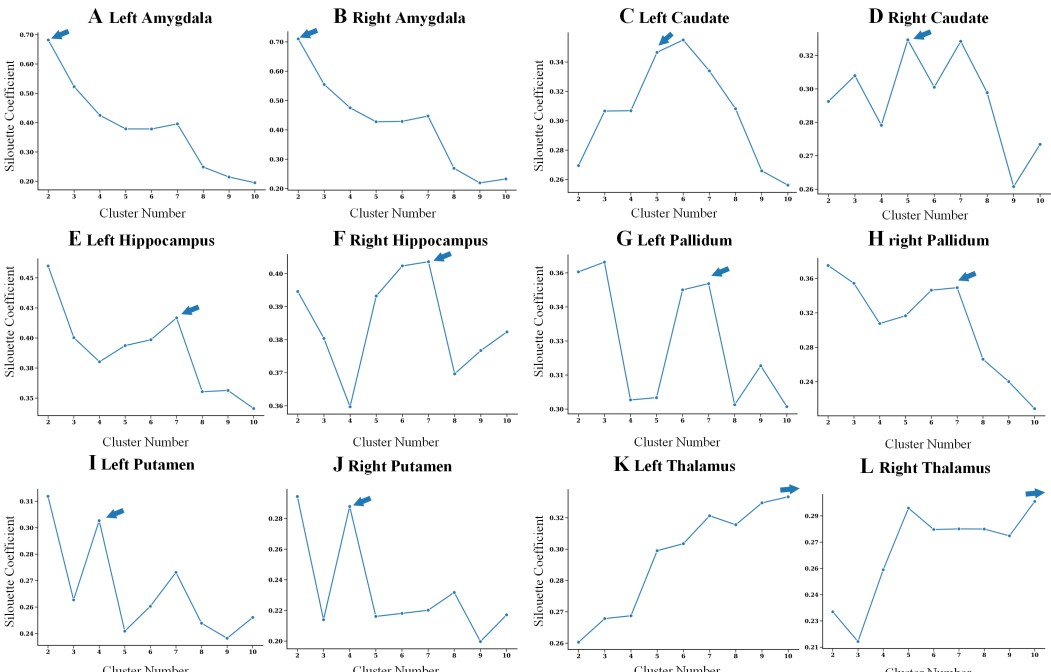

Figure 7: Silhouette coefficient optimization profiles for subcortical nucleus parcellation. SC values are plotted across subregion numbers $r \in \{2, 3, \ldots, 10\}$ for five nuclei (amygdala, caudate nucleus, hippocampus, pallidum, putamen) and extended to $r \in \{2, 3, \ldots, 15\}$ for thalamus due to its complex internal architecture. Arrows denote selected optimal configurations determined through combined SC maximization and neuroanatomical validation. The analysis demonstrates nucleus-specific structural complexity gradients, with optimal parcellation granularities established as follows: amygdala ($r = 2$), caudate nucleus ($r = 5$), hippocampus ($r = 7$), pallidum ($r = 2$), putamen ($r = 4$), and thalamus ($r = 10$).

subregion numbers for subsequent neuroanatomical characterization: amygdala ($r = 2$), caudate nucleus ($r = 5$), hippocampus ($r = 7$), pallidum ($r = 2$), putamen ($r = 4$), and thalamus ($r = 10$).

## D.5 PARAMETER SENSITIVITY ANALYSIS

To further assess the model's parameter sensitivity, we conducted a fine-grained analysis centered around the previously identified favorable settings. We systematically adjusted the regularization weights, evaluating 10 distinct values for each parameter. Specifically, $\lambda_1$ and $\lambda_2$ were varied within the range of [0.01, 0.10] using a step size of 0.01, while $\lambda_3$ was adjusted within [0.05, 0.50] using a step size of 0.05. All other experimental conditions remained consistent with those detailed in Section B.3. The evaluation was based on the mean silhouette coefficient (SC) averaged across 30 independent runs on the bilateral thalamus. As shown in Table 9, the results demonstrate that per-

formance metrics remain highly stable within this local parameter range, confirming the robustness of our selected configuration.

### D.6 COMPARISON OF SYMMETRY-PRESERVING STRATEGIES

To further validate the effectiveness of our symmetry constraint, we conducted additional ablation experiments comparing two alternative symmetry-preserving strategies for subcortical parcellation. The first strategy, termed input-level symmetrization, computes a weighted average of left and right hemisphere data followed by mirror symmetry prior to training. This approach aligns with the methodology in Tian et al. (2020), which directly mirrors left-right hemisphere data prior to clustering. The second strategy, termed output-level symmetrization, first trains a single-path USDPnet independently on each hemisphere and performs clustering, then averages the resulting soft assignment matrices $A$ across symmetric correspondences established using the same matching procedure described in the main text, yielding fully mirrored bilateral parcellation results. As shown in Table 4, using identical hyperparameters and 30 random seeds as in the main experiments, our method outperforms both alternatives across all four metrics (SC/CH/RE/FH). For fair comparison, all evaluations are conducted on the original features. Notably, our approach enforces soft symmetry during training by minimizing the discrepancy between $A_L$ and $A_R$, rather than imposing hard constraints on the final parcellation, thus preserving feature-driven asymmetries as the divergence loss dominates over the symmetry loss.

Table 9: Fine-grained sensitivity analysis for regularization parameters $\lambda_1$, $\lambda_2$, and $\lambda_3$. Parameters $\lambda_1$ and $\lambda_2$ were varied from 0.01 to 0.10 with a step size of 0.01, while $\lambda_3$ was varied from 0.05 to 0.50 with a step size of 0.05. Performance is measured by the Silhouette Coefficient on the left and right thalamus.

| Parameters $\lambda_1$ and $\lambda_2$ (Range: 0.01-0.10, Step: 0.01) | | | | | | | | | | |
|---|---|---|---|---|---|---|---|---|---|---|
| **Parameter** | **0.01** | **0.02** | **0.03** | **0.04** | **0.05** | **0.06** | **0.07** | **0.08** | **0.09** | **0.10** |
| $\lambda_1$ (left) | 0.221 | 0.235 | 0.228 | 0.245 | 0.248 | **0.251** | 0.246 | 0.239 | 0.232 | 0.231 |
| $\lambda_1$ (right) | 0.200 | 0.215 | 0.229 | **0.249** | 0.248 | 0.244 | 0.245 | 0.238 | 0.225 | 0.223 |
| $\lambda_2$ (left) | 0.185 | 0.208 | 0.215 | 0.218 | 0.232 | **0.244** | 0.231 | 0.227 | 0.219 | 0.212 |
| $\lambda_2$ (right) | 0.194 | 0.212 | 0.218 | 0.235 | 0.242 | **0.251** | 0.235 | 0.222 | 0.219 | 0.213 |
| Parameter $\lambda_3$ (Range: 0.05-0.50, Step: 0.05) | | | | | | | | | | |
| **Parameter** | **0.05** | **0.10** | **0.15** | **0.20** | **0.25** | **0.30** | **0.35** | **0.40** | **0.45** | **0.50** |
| $\lambda_3$ (left) | 0.191 | 0.253 | **0.258** | 0.249 | 0.235 | 0.244 | 0.215 | 0.192 | 0.158 | 0.125 |
| $\lambda_3$ (right) | 0.205 | **0.255** | 0.253 | 0.251 | 0.233 | 0.235 | 0.221 | 0.188 | 0.165 | 0.126 |

## E EXPERIMENTAL DATA AND PREPROCESSING PIPELINE

### E.1 BCP DATASET CHARACTERISTICS

The Baby Connectome Project (BCP) dataset (Howell et al., 2019) provides longitudinal neuroimaging data from typically developing infants. For each scan, both T1w and T2w images were collected with 3T Siemens Prisma MRI scanners using a 32-channel head coil. The T1w images were acquired with parameters: TR/TE/TI = 2400/2.24/1060 ms, flip angle = 8°, and isotropic spatial resolution of 0.8 mm. The T2w images were acquired with parameters: TR/TE = 3200/564 ms, variable flip angle, and isotropic spatial resolution of 0.8 mm. Table 10 presents the details of the data used in this work. Of note, the BCP cohort doesn't include twins. Figure 8 exhibits the number of subjects per month with respect to sex (Female/Red, Male/Blue).Additional methodological details can be found in (Howell et al., 2019).

**Inclusion and Exclusion Criteria.** *Inclusion criteria:* (1) born from 37 to 42 weeks gestational age (GA); (2) appropriate birth weight matching gestational age; (3) absence of major pregnancy and delivery complications.*Exclusion criteria:* (1) adoption status; (2) diagnosed schizophrenia, autism

Table 10: Participant characteristics and data processing workflow for the BCP dataset.

| Characteristic | Data |
|---|---|
| Total number of scans acquired | 702 scans |
| **Remaining scans after each processing step:** | |
|    1. After quality control (excessive motion, insufficient coverage, and/or ghosting) | 633 scans |
|    2. After removing scans with missing T1w or T2w images | 564 scans |
|    3. After subcortical segmentation and manual correction | 513 scans |
| **Total number of subjects** | 231 subjects (513 scans) |
|    Subjects having only 1 scan | 92 subjects |
|    Subjects having 2 scans | 65 subjects |
|    Subjects having $\geq 3$ scans | 74 subjects |
| Sex distribution | 126 females / 105 males |
| Age range | 10–809 days (scan age) |

Figure 8: The number of subjects at each scan age.

spectrum disorder, bipolar disorder, or intellectual disability; (3) medical or genetic conditions affecting growth, development, or cognition; (4) MRI contraindications; (5) maternal substance use (alcohol or illicit drugs), placental abruption, maternal preeclampsia, or maternal HIV-positive status during pregnancy.

### E.2 NEUROIMAGING PREPROCESSING METHODOLOGY

We strictly adhered to the comprehensive iBEAT V2.0 deep learning–based pipeline as detailed in (Wang et al., 2023) for all infant cortical surface reconstruction procedures.

**Subcortical Surface Mapping Protocol.** Instead of reconstructing any new atlas, we followed and applied the publicly released 4D subcortical atlas and mapping framework described in Chen et al. (2022), and we include the workflow here solely to document the preprocessing methodology used in our study. Figure 9 illustrates the complete subcortical surface mapping workflow. The processing pipeline comprises five sequential stages:

*(1) 4D Atlas Construction:* The publicly available 4D infant brain atlas Chen et al. (2022) is utilized and processed through the SyGN template construction method in ANTs. T1w, T2w, and tissue

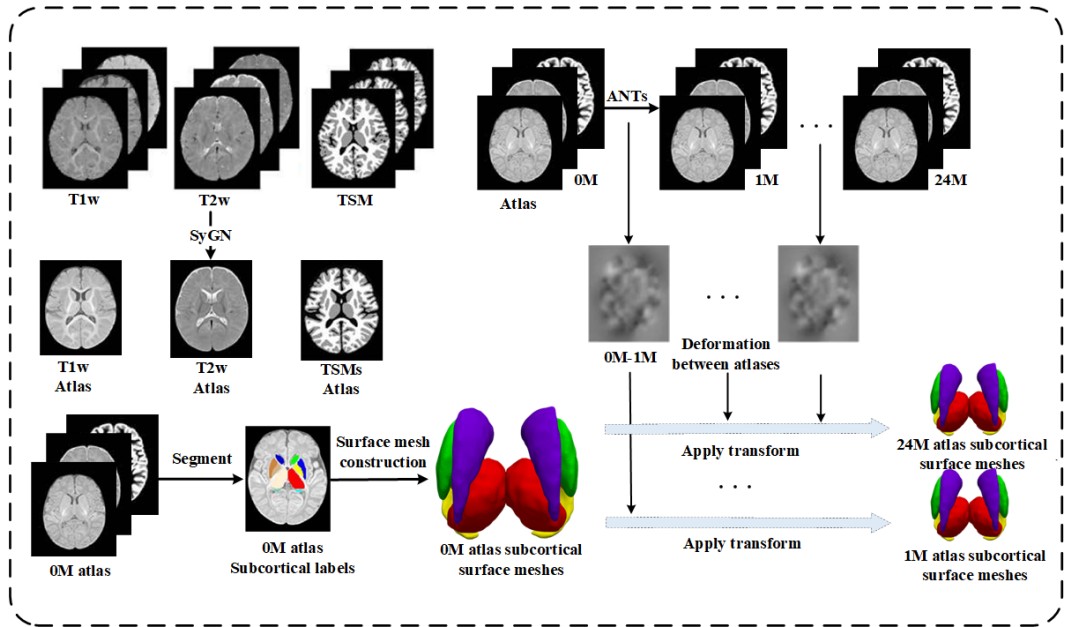

Figure 9: Overview of the cortical surface reconstruction pipeline. M: Month. TSM: tissue segmentation map.The workflow illustrates the complete processing pipeline from multi-modal MRI data (T1w, T2w, TSM) through atlas-based registration and deformation mapping to generate subcortical surface meshes across different developmental stages. ANTs registration enables transformation propagation from atlas to individual subjects, facilitating vertex-correspondence analysis of subcortical nuclei morphometry.

probability maps (generated by iBEAT V2.0) are incorporated to improve registration robustness across developmental stages. The atlas includes densely sampled temporal points from 0 to 24 months (0, 1, 2, 3, 4, 5, 6, 7, 8, 9, 10, 11, 12, 15, 18, 21, 24 months). Age-specific deformation fields provided in the framework are used for linking individual scans to atlas templates across time.

*(2) Inter-Atlas Deformation Mapping:* Voxel-wise anatomical deformations across developmental time points are computed following the procedures described in Chen et al. (2022).High-contrast images serve as registration targets, with low-contrast images warped accordingly.Temporal consistency is maintained through sequential chronological registration.

*(3) Reference Surface Reconstruction:* Surface mesh representations were reconstructed for each subcortical structure using the 0-month atlas as the initial reference template.

*(4) Individual Surface Warping:* Subcortical surface meshes from the 0-month template were warped to individual scans by combining anatomical deformations across age-specific atlases with template-to-individual transformations in chronological sequence. These deformation fields were consolidated into unified transformations for efficient surface mapping.

*(5) Vertex-wise Feature Extraction:* Local surface areas were computed at each vertex across individual subcortical surfaces to establish vertex-wise developmental trajectories.

The integration of tissue probability maps as additional registration constraints effectively addresses dynamic appearance changes and low tissue contrast characteristic of infant brain MRI, resulting in accurate deformation estimation and high-quality subcortical surface correspondence across developmental stages.

### E.3 SUBCORTICAL SEGMENTATION AND MANUAL CORRECTION

Subcortical segmentation is based on an infant-dedicated deep learning framework (Chen et al., 2020; 2023b).This section summarizes the segmentation and manual-correction steps below to provide context for the preprocessing workflow adopted in our study. In detail, at the coarse stage, the

pipeline uses the SDM-UNet(Chen et al., 2020) to directly predict the signed distance maps from multi-modal intensity images, including T1w, T2w, and the ratio of T1w and T2w images, which can leverage the spatial context information, including the structural position information and the shape information of the target structure, to generate high-quality signed distance maps. At the fine stage, the pipeline further uses a multi-source and multi-path attention UNet (M2A-UNet)(Chen et al., 2023b). Then, the signed distance maps predicted by SDM-UNet, which encode spatial-context information of each subcortical structure, are integrated with the multi-modal intensity images as the input of M2A-UNet for achieving refined segmentation. Besides, both the 3D spatial and channel attention blocks are added to guide the M2A-UNet to focus more on the important subregions and channels. Due to the significantly different appearances of the infant brain MR images across ages, we manually delineated 48 scans within four representative age ranges, namely 0-3 months, 6 months, 9-12 months, and 18-24 months, and each age range has 12 scans. We then separately trained a deep network for each age group. A stratified 6-fold cross-validation strategy is employed, and each fold consists of 10 training images and 2 testing images.

## F  MULTI-PERSPECTIVE PARCELLATION VISUALIZATION

This section presents comprehensive three-dimensional visualization of subcortical nuclei parcellation outcomes across multiple methodological approaches. Figure 10 demonstrates the parcellation results obtained through our proposed USDPnet framework, with subcortical structures rendered in bilateral paired configurations to facilitate systematic morphological assessment. Each nucleus pair is visualized from six distinct anatomical perspectives to enable comprehensive evaluation of three-dimensional parcellation boundaries and subregional organization patterns.

Figures 11–14 provide equivalent multi-perspective renderings for baseline methodologies: Spectral clustering with discretization (top-performing spectral clustering variant), DeepNMF (established matrix factorization approach), DDC (recent deep clustering method), and Tian et al. (2020)'s functional connectivity gradient-based parcellation using voxel data. Visual inspection reveals several key advantages of our USDPnet framework over these baseline approaches. The proposed method demonstrates superior boundary definition precision, with more anatomically coherent subregional delineations that preserve neurobiologically meaningful structures. Additionally, dynamic visualizations of our parcellation results can be viewed in the supplementary movie.mp4 and movie1.mp4 files.

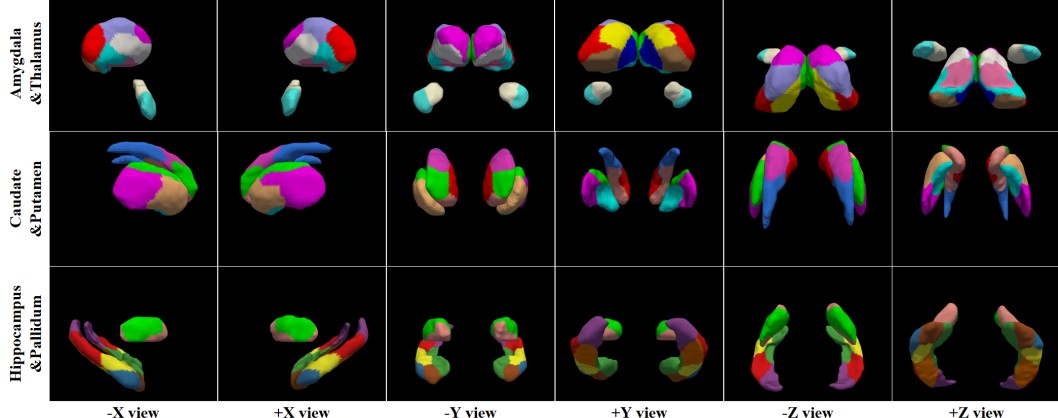

Figure 10: Subcortical nuclei parcellation results obtained through the proposed USDPnet framework, demonstrating multi-perspective anatomical renderings with bilateral paired configurations for comprehensive morphological assessment.

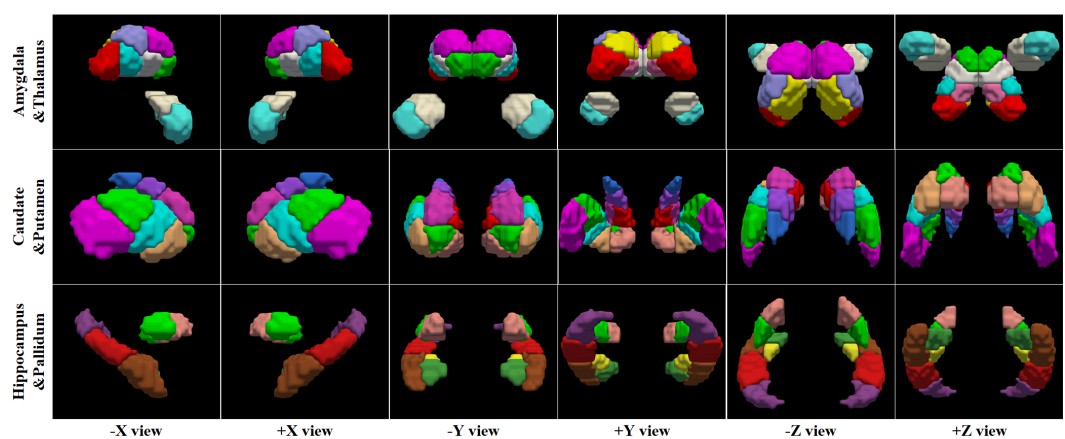

Figure 11: Subcortical nuclei parcellation outcomes obtained via Tian et al. (2020)'s functional connectivity gradient-based approach.

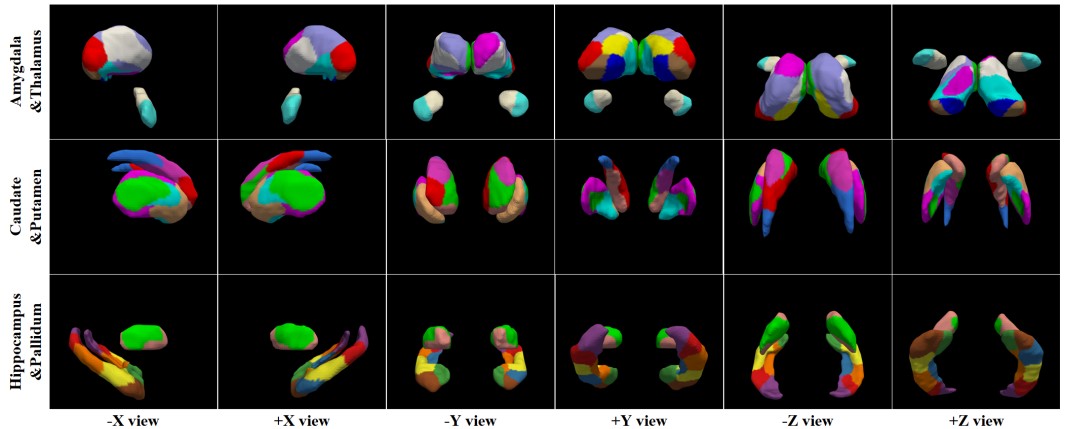

Figure 12: Subcortical nuclei parcellation outcomes obtained via DDC methodology.

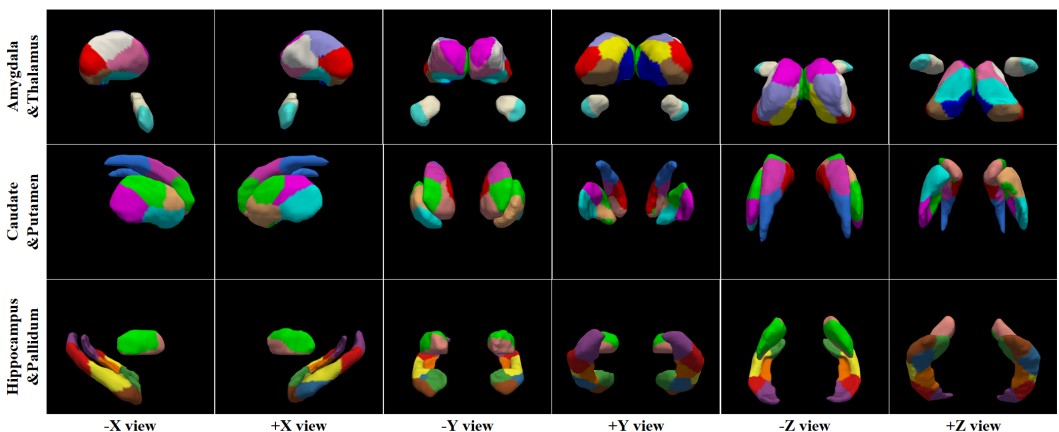

Figure 13: Subcortical nuclei parcellation outcomes obtained via DeepNMF methodology.

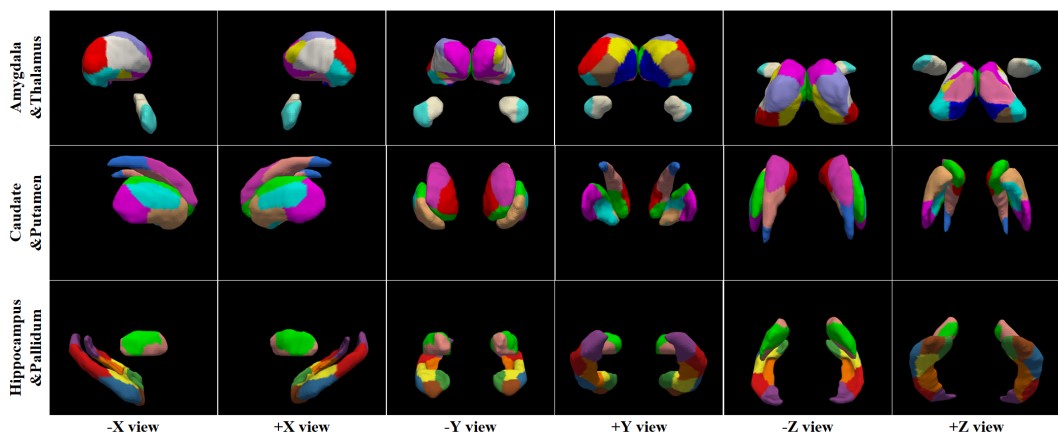

Figure 14: Subcortical nuclei parcellation outcomes obtained via Spectral clustering with discretization methodology.

