# OpenReview forum: "USDPnet: An Unsupervised Symmetric Deep Framework for Robust Parcellation of Infant Subcortical Nuclei"
_ICLR.cc/2026/Conference — Submitted to ICLR 2026_

### Official Review · Reviewer_xuse · 2025-10-28

**Soundness:** 3
**Presentation:** 4
**Contribution:** 3
**Rating:** 6
**Confidence:** 4

**Summary:**

The study imposes mirror symmetry in parcellation of infant subcortical nuclei, that is known to be structurally symmetrical and processed with methods for adult brain parcellation methods.
The method depends on an autoencoder-based architecture that process the left and right hemisphere vertices with a  symmetry-aware clustering mechanism that utilizes Generalized Cauchy-Schwarz divergence on surface meshes. The clustering loss favors bilateral consistency, higher inter-cluster sample distances, lower intra-cluster sample distances and sparse cluster assignment vectors.
The results provide a thorough ablation and various baseline clustering methods, showing the proposed method has higher performance than the baselines.

**Strengths:**

- The bilateral symmetry-aware segmentation/parcellation is a recent topic [1, 2] and the study suggests a sound method for the problem.
- The divergence measure Generalized Cauchy-Schwarz Distribution introduces computational improvements over average-pairwise divergence measures.
- The study shows strong performance against alternative clustering baselines.

**Weaknesses:**

- Lack of Ablation on Core Component: The paper's claim of a novel divergence function is insufficiently supported, as the ablation study does not compare against alternative divergence measures (e.g., KL-divergence, JS-divergence). Without this comparison, it is impossible to assess whether the proposed function is truly responsible for the performance gains or if other architectural choices are the primary driver.
- References to other bilateral symmetry-aware segmentation/parcellation: The study can involve alternatives in different areas of research, i.e. [1,2].

[1] Sanket Wathore, Subrahmanyam Gorthi, Bilateral symmetry-based augmentation method for improved tooth segmentation in panoramic X-rays, Pattern Recognition Letters, Volume 188, 2025, Pages 1-7, ISSN 0167-8655, https://doi.org/10.1016/j.patrec.2024.11.023.

[2] Raina, K., Yahorau, U. and Schmah, T. Exploiting Bilateral Symmetry in Brain Lesion Segmentation with Reflective Registration.
DOI: 10.5220/0008912101160122, In Proceedings of the 13th International Joint Conference on Biomedical Engineering Systems and Technologies (BIOSTEC 2020) - Volume 2: BIOIMAGING, pages 116-122, ISBN: 978-989-758-398-8; ISSN: 2184-4305

Typo: Line 456-457 vactors -> vectors

**Questions:**

Could the authors comment on the performance of the framework when using divergence measure alternatives to $D_{GCS}$, other than GJRD, beyond the computational advantages? Specifically, if the components A and Q are kept intact, how do alternative measures (average-pairwise divergences) compare in terms of both performance and the computational advantages highlighted in the paper?

[1] Mingfei Lu, Lei Xing, Badong Chen,
Measuring generalized divergence for multiple distributions with application to deep clustering,
Pattern Recognition,
Volume 157,
2025,
110864,
ISSN 0031-3203,
https://doi.org/10.1016/j.patcog.2024.110864.

---

> ### Author Response · Authors · 2025-11-22
> **Responses to reviewer xuse -A [1/2]**
>
> # Reviewer xuse
>
> Thank you for your constructive feedback and thoughtful suggestions. Our point-by-point responses are below.
>
> ## Response to Weaknesses and Questions
>
> ### Weakness 1, Question 1
>
> **Reviewer's Question:** "Lack of Ablation on Core Component."
>
> **Response:** We thank the reviewer for this crucial observation. To more clearly demonstrate the advantages of our divergence compared with alternative choices, we conducted a direct ablation study. Specifically, we compared our method against **Cauchy-Schwarz divergence (CSD)**, a pairwise variant of our proposed measure; **Kullback-Leibler divergence (KLD)**, a standard information-theoretic measure; and **Generalized Jensen-Rényi divergence (GJRD)**, specifically GJRD-1 and GJRD-2, which correspond to Rényi orders 1 (reducing to Jensen–Shannon divergence, **JSD**) and 2 (reducing to Rényi entropy of order 2), respectively.
>
> For pairwise comparisons, we selected the best-performing results across multiple runs (By convention, clustering tasks are usually evaluated using the best-performing result across runs). We evaluated performance using four metrics: Silhouette Coefficient (SC), Calinski-Harabasz Index (CH), Reconstruction Error (RE), and Feature Homogeneity (FH). These results have been incorporated into the new **"Comparison of different divergences" section on Page 10** and are summarized in **Table 3** below.
>
> | Method | SC (L/R) ↑ | CH (L/R) ↑ | RE (L/R) ↓ | FH (L/R) ↑ |
> | :--- | :---: | :---: | :---: | :---: |
> | CSD | 0.252 / 0.255 | 1075 / 1003 | 1.149 / 1.167 | 0.689 / 0.678 |
> | KLD | 0.218 / 0.200 | 1056 / 1007 | 1.149 / 1.145 | 0.680 / 0.676 |
> | GJRD-1 (JSD) | 0.281 / 0.274 | 965 / 865 | 1.212 / 1.255 | 0.664 / 0.653 |
> | GJRD-2 | 0.264 / 0.265 | 970 / 895 | 1.208 / 1.228 | 0.668 / 0.656 |
> | **Ours (GCSD)** | **0.328 / 0.317** | **1228 / 1123** | **1.042 / 1.087** | **0.708 / 0.704** |
>
> Across all configurations, GCSD achieves the most favorable balance between clustering quality, robustness to outliers, and computational efficiency. Unlike pairwise divergences (CSD, KLD) that scale as $O(r^2)$ for $r$ clusters, GCSD enables joint optimization across all clusters simultaneously ($O(r)$). This makes it particularly well-suited for processing noisy, highly variable infant brain imaging data and for high-granularity parcellation tasks (e.g., the thalamus with 10 subregions).

---

> ### Author Response · Authors · 2025-11-22
> **Responses to reviewer xuse -A [2/2]**
>
> ### Weakness 2
>
> **Reviewer's Question:** "References to Other Works."
>
> **Response:** We appreciate the reviewer bringing attention to recent work on bilateral symmetry in medical imaging. We have ensured that these references [1, 2] are cited on **page 2, line 83** of our revised manuscript, where we discuss the importance of exploiting bilateral symmetry.
>
> *   **Wathore et al. [1]** demonstrate the effectiveness of bilateral symmetry-based **data augmentation** for tooth segmentation. Their approach exploits inherent symmetry to increase training data, leading to substantial improvements in supervised models (e.g., TransUNet). This validates the general principle that leveraging symmetry enhances model performance.
> *   **Raina et al. [2]** exploit bilateral quasi-symmetry for brain lesion segmentation through **feature augmentation** (reflective registration). By adding homologous voxel patches as features, they achieve significant improvements. This demonstrates the value of symmetry constraints even when anatomical symmetry is affected by pathology.
>
> While these studies utilize symmetry for *data* or *feature augmentation* in supervised settings, our work incorporates bilateral symmetry as an **explicit regularization constraint ($\mathcal{L}_{\text{sym}}$)** within an *unsupervised* clustering framework. The concrete technical pipelines in these two papers crucially rely on the combination of “labels + voxel/pixel space + segmentation loss”: Raina’s construction of mirrored patch features must operate in the full-brain 3D intensity domain and optimize a CNN around lesion labels, whereas USDPnet performs clustering only on reconstructed subcortical surfaces and high-dimensional developmental trajectories; Wathore’s symmetry-based augmentation, in turn, requires accurate instance-level tooth segmentations and numbering before flipping and re-labeling in 2D images, while USDPnet neither has such instance-level annotations nor is it appropriate to perform large geometric flips and re-numbering on already aligned surface meshes.
>
> In other words, these two works use symmetry to enhance supervised segmentation and thus conceptually support the idea that “symmetry information is valuable,” but their task objectives, data modalities, and loss designs are all incompatible with the unsupervised symmetry-based clustering framework of USDPnet, and therefore they cannot be directly adopted as alternative methods or baselines in our setting.
>
> ### Minor Correction
>
> "Typo: Line 456-457 vactors -> vectors":Thank you for catching this typo. It has been corrected in the revised version on line 477.

---

> ### Comment · Reviewer_xuse · 2025-11-27
>
> Thank you for the comprehensive rebuttal and for conducting the new ablation study. This addresses my concern by providing empirical support for the adoption of GCSD over other divergence measures. I agree that the results in the new table convincingly demonstrate the benefits of GCSD in terms of clustering quality and reconstruction fidelity on the given dataset.
>
> However, two key points require further clarification to fully substantiate the paper's claims:
>
>    * Claim of "Robustness to Outliers" (Weakness 1): The assertion that GCSD provides "robustness to outliers" remains implied rather than directly validated. The presented metrics (SC, CH, RE, FH) measure clustering quality on the pristine dataset but do not constitute a test of robustness against noise or anomalies. To fully support this claim, it would be necessary to either:
>
>        * conduct experiments with synthetically injected outliers or noise, or
>        * evaluate on data with known outlier samples, or
>        * state that GCSD achieves superior clustering quality and computational efficiency, which may contribute to robustness—a hypothesis for future work.
>
>    * Substantiation of the Symmetry Constraint (Weakness 2): My core methodological concern in suggesting the alternatives is due to the lack of rigorous testing of the symmetry constraint. By directly penalizing inter-hemispheric differences, the constraint artificially reduces sample variance. This very likely inflates cluster quality metrics, which intrinsically reward such homogeneity. This effect may explain why the ablation of the symmetry term (w/o L_sym) causes the most significant performance drop.
>
> While leveraging the known symmetry of infant brain structure is a valid motivation, the current evaluation cannot distinguish between the method's ability to discover latent symmetry versus enforcing it to optimize metrics. To better substantiate the novelty and necessity of the proposed joint optimization framework, a comparison against conceptually simpler symmetry-preserving alternatives is needed. For instance, could a post-hoc symmetrization of independently clustered hemispheres achieve similar gains? Such an ablation would demonstrate the unique advantage of the proposed in-training symmetry regularization.
>
> In summary, while the GCSD ablation is compelling, the paper would be significantly strengthened by addressing the "robustness" claim and providing a more rigorous validation of the symmetry constraint against simpler conceptual alternatives.

---

> > ### Author Response · Authors · 2025-12-03
> > **Responses to reviewer xuse -B**
> >
> > # Response to Reviewer xuse
> >
> > Thank you for your prompt follow-up and constructive suggestions.
> >
> > ## Weakness 1: Claim of "Robustness to Outliers"
> >
> > We sincerely thank the reviewer for this insightful observation. Upon careful reflection, we acknowledge that our current experimental evidence does not directly validate robustness against noise or anomalies. The metrics we employed (SC, CH, RE, FH) primarily assess clustering quality on the original dataset rather than explicitly testing outlier resilience.
> >
> > We have revised the manuscript to remove the unsupported claim of "robustness to outliers." Instead, we now state that GCSD achieves superior clustering quality and computational efficiency, which *may* contribute to robustness—a hypothesis we leave for future investigation. Specifically, we plan to conduct experiments with synthetically injected outliers or evaluate on data with known anomalous samples in future work to rigorously validate this property.
> >
> > We appreciate the reviewer's guidance in helping us present our contributions more accurately.
> >
> > ## Weakness 2: Substantiation of the Symmetry Constraint
> >
> > We thank the reviewer for raising this important methodological concern. We address this from both theoretical and empirical perspectives.
> >
> >
> > We appreciate the reviewer's concern that penalizing inter-hemispheric differences may artificially reduce sample variance and inflate cluster quality metrics. We would like to offer the following clarifications: (1) our symmetry loss operates on soft assignment matrices $A_L$ and $A_R$, rather than on the original features $X$ where clustering metrics are computed; (2) the symmetry weight $\lambda_3 = 0.1$ is relatively small compared to the GCSD terms, allowing cluster assignments to be primarily driven by data structure; and (3) unlike hard constraints, our soft symmetry only *encourages* bilateral consistency while preserving feature-driven asymmetries.
> >
> > To directly address the reviewer's suggestion, we conducted additional ablation experiments (**Appendix D.6**) comparing our in-training symmetry regularization against two conceptually simpler alternatives. The first strategy, *input-level symmetrization*, computes a weighted average of left and right hemisphere data followed by mirror symmetry prior to training, similar to the approach in Tian et al. (2020), which directly mirrors bilateral data before clustering. The second strategy, *output-level symmetrization*, first trains a single-path USDPnet independently on each hemisphere and performs clustering, then averages the resulting soft assignment matrices $A$ across symmetric correspondences, yielding fully mirrored bilateral parcellation results. Notably, both alternatives enforce *harder* symmetry constraints than our method—pre-training symmetry homogenizes the input features entirely, while post-training symmetry forces identical final parcellations. If the reviewer's hypothesis were correct (that symmetry enforcement inflates metrics), these alternatives should achieve higher or comparable scores. However, using identical hyperparameters and 30 random seeds as in the main experiments, and evaluating all methods on the original features for fair comparison, our method outperforms both alternatives across all four metrics:
> >
> > |Symmetrization Strategy | SC (L/R) ↑ | CH (L/R) ↑ | RE (L/R) ↓ | FH (L/R) ↑ |
> > |----------|------------|------------|------------|------------|
> > | Input-level  | 0.303 / 0.302 | 1129 / 1130 | 1.116 / 1.116 | 0.686 / 0.688 |
> > | Output-level | 0.288 / 0.289 | 1119 / 1119 | 1.212 / 1.211 | 0.663 / 0.664 |
> > | **Ours(In-training)** | **0.328 / 0.317** | **1228 / 1123** | **1.042 / 1.087** | **0.708 / 0.704** |
> >
> > These results demonstrate that our in-training soft symmetry regularization provides a unique advantage over simpler alternatives, and that harder symmetry constraints actually *degrade* performance—supporting our claim that the symmetry loss discovers latent bilateral structure rather than merely enforcing homogeneity to inflate metrics.

---

### Official Review · Reviewer_Tsms · 2025-10-31

**Soundness:** 2
**Presentation:** 3
**Contribution:** 2
**Rating:** 4
**Confidence:** 4

**Summary:**

This paper proposes USDPnet, an unsupervised deep clustering framework incorporating anatomical symmetry constraints for fine-grained infant subcortical parcellation. The method leverages surface-mesh vertex area trajectories, a latent representation encoder, and a Generalized Cauchy–Schwarz Divergence (GCSD) objective, along with a hemisphere-pairing MSE symmetry loss.
Experiments on the Baby Connectome Project (BCP) dataset demonstrate improvements over several conventional and deep clustering baselines, accompanied by ablation studies and statistical significance testing. Visualizations indicate anatomically reasonable results and improved bilateral consistency.

**Strengths:**

1. The paper tackles the important and challenging problem of infant subcortical parcellation, which is clinically relevant and understudied in the unsupervised setting.
2. The approach is label-free and has potential value for large-scale infant neurodevelopment studies where manual annotations are difficult or costly to obtain.
3. The manuscript is generally well-organized, with clear presentation of the model design, loss components, and visual examples.

**Weaknesses:**

1. Lack of external validation with anatomical ground truth

    No Dice, ARI, or NMI comparison to expert labels or standard infant atlases. Reliance on internal clustering metrics limits biological interpretability.

2. Risk of suppressing true biological asymmetry

    The symmetry constraint may over-regularize regions with known asymmetries (e.g., amygdala, thalamus). No analysis provided to quantify the impact or demonstrate robustness.

3. Scalability concerns

    The full-batch GCSD objective may not scale to larger datasets or higher-resolution surfaces. No discussion on computational efficiency or potential approximations.

4. Limited feature modalities

    Only vertex-area trajectories are used. Incorporating curvature, thickness, deformation tensors, or multimodal T1/T2 contrast might provide more stable clustering.

5. FH metric insufficiently defined

    The proposed Feature Homogeneity metric is not formally introduced in the main text, limiting reproducibility and interpretation.

6. Recent literature coverage is insufficient

    Related work on deep clustering and infant brain parcellation from the past 3 years is under-represented.

**Questions:**

1. The authors cite Lu et al. (2025b) for the GCSD estimator. Please clarify the precise differences between that prior work and the current implementation — specifically, have you introduced any modifications or simplifications relative to the original estimator?

2. You note that small asymmetries may lead to misleading conclusions. The manuscript also demonstrates to some extent that imposing a symmetry constraint aids segmentation. However, what happens if the input data include small but true asymmetries? Can the proposed method maintain robustness under such conditions, or does the symmetry‐constraint unduly suppress biologically meaningful asymmetry?

3. The use of MSE as the loss for the symmetry constraint raises a concern: might this penalty unintentionally penalize genuine developmental asymmetries? Have you considered employing a weighted or soft symmetry constraint (for example, applying it only to high‐confidence regions) to avoid suppressing valid anatomical differences?

4. The metric “Feature Homogeneity” (FH) appears in your results, but I could not find its formal definition in the main text. Please provide in the rebuttal the exact formula for FH and clarify its physical/biological interpretation.

5. In the figures and tables, please ensure that appropriate legends and annotations are included so that readers can understand the visualized results without excessive ambiguity.

6. Regarding GCSD computation, Equation (2) involves logarithms and matrix products, which may be prone to numerical instability. Could you clarify how potential underflow or negative values are handled to ensure safe log computation?

7. The current reference list seems to under‑represent the past three years of literature in deep clustering and infant brain parcellation. Please consider incorporating more recent studies to demonstrate how your work builds upon and differs from the state‑of‑the‑art.

---

> ### Author Response · Authors · 2025-11-22
> **Responses to reviewer Tsms -A [1/3]**
>
> ## Reviewer Tsms
> Thank you for your constructive comments. Our point-by-point responses are below.
>
> ## Response to Weaknesses and Questions
>
> ### Weakness 1
>
> **Reviewer's Question:** "External validation."
>
> **Response:** We acknowledge the value of validation against anatomical ground truth. However, its acquisition for infant (0-24 months) subcortical parcellation is infeasible due to fundamental challenges: low tissue contrast and rapid morphological changes make expert annotation unreliable. Critically, no infant atlas with subregional parcellations exists to serve as ground truth; existing atlases are derived from adult data and are unsuitable.
>
> To validate our approach, we first confirmed its **reproducibility and stability** by running our method 30 times and computing Dice, ARI, and NMI across these runs, as well as by visually comparing the resulting parcels with adult atlases to assess their anatomical plausibility. These analyses demonstrate that our method yields highly stable clustering solutions and parcels that closely correspond to known adult subcortical subdivisions. The quantitative results are presented on **Page 6 (\*Competing Methods and Metrics\*)** and in **Table 1**. In the absence of direct ground truth, this strong reproducibility and anatomical consistency provide the most convincing validation currently possible, which also supports our use of **unsupervised methods as the only viable option** in this setting.
>
> |Method|Dice|ARI|NMI|
> |-|-|-|-|
> |DEC|0.51 ± 0.06|0.30 ± 0.05|0.37 ± 0.09|
> |DDC|0.54 ± 0.08|0.35 ± 0.04|0.44 ± 0.10|
> |GJRD|0.60 ± 0.04|0.41 ± 0.07|0.47 ± 0.12|
> |**Ours**|**0.65 ± 0.02**|**0.47 ± 0.05**|**0.57 ± 0.11**|
>
> These results, together with the table below, show that our method is more stable and accurate than the baselines.
>
> ## Weakness 2, Question 2-3
>
> **Reviewer's Question:** "The symmetry constraint."
>
> **Response:** This is an excellent point regarding the risk of suppressing true biological asymmetry. Our MSE-based symmetry loss is intentionally designed as a **soft constraint** that encourages, but does not enforce, perfect bilateral correspondence. Because the symmetry loss weight is small, the impact of inherent data asymmetries on the primary clustering loss is far greater, allowing local asymmetries to emerge naturally.
>
> In contrast to methods like Tian et al. [1], which enforce feature mirroring at the data processing stage to improve SNR and stability, our approach preserves local asymmetries, leading to more interpretable results while maintaining robust performance.
>
> To directly address whether a weighted or softer constraint might better preserve valid anatomical differences, we conducted systematic experiments evaluating this very idea. As detailed in **Appendix D.1**, we compared our fixed-weight strategy against three scheduling strategies (cosine, linear, and exponential) that progressively reduce the symmetry weight during training, effectively implementing a dynamic 'soft' constraint.
>
> | Scheduling Strategy | SC (L/R) ↑ | CH (L/R) ↑ | RE (L/R) ↓ | FH (L/R) ↑ |
> | :--- | :---: | :---: | :---: | :---: |
> | Cosine | 0.324 / 0.320 | 1211 / 1120 | 1.038 / 1.085 | 0.696 / 0.706 |
> | Linear | 0.315 / 0.311 | 1154 / 1146 | 1.102 / 1.110 | 0.692 / 0.688 |
> | Exponential | 0.311 / 0.309 | 1142 / 1055 | 1.112 / 1.119 | 0.686 / 0.679 |
> | **Ours (Fixed)** | **0.328 / 0.317** | **1228 / 1123** | **1.042 / 1.087** | **0.708 / 0.704** |
>
> The results surprisingly show that dynamically reducing the symmetry weight consistently **degrades** performance across all metrics. The cosine, linear, and exponential decay strategies led to performance drops of up to 1.2%, 4.0%, and 5.2% in the SC metric, respectively.
>
> This analysis indicates that our fixed-weight strategy is well justified. Compared with the hard mirror-symmetry enforcement used in Tian et al. [1], our approach better preserves the global bilateral organization while accommodating localized, neurobiologically meaningful asymmetries. These results further suggest that, for this task, adopting a fixed soft constraint is the optimal strategy.
>
> [1] Tian, Ye, et al. "Topographic organization of the human subcortex unveiled with functional connectivity gradients." Nature Neuroscience 23.11 (2020): 1421-1432.

---

> ### Author Response · Authors · 2025-11-22
> **Responses to reviewer Tsms -A [2/3]**
>
> ### Weakness 3
>
> **Reviewer's Question:** "Scalability of full-batch strategy."
>
> **Response:** Our full-batch strategy is both computationally efficient and critical for performance. As shown in **Appendix B.4**, training completes in minutes on a single NVIDIA RTX 3090 GPU, with memory scaling linearly and peaking at just 1.80 GB for the thalamus. This demonstrates the method is highly practical for real-world applications.
>
> To further validate this, we systematically compared full-batch training against mini-batch strategies (batch sizes: 64, 256, 1024)，As shown in **Appendix D.2**. Experiments on the thalamus (3,416/3,346 vertices, L/R), the structure with the highest vertex count, for $r=10$ subregions show that mini-batching significantly degrades performance.
>
> | Batch Strategy | SC (L/R) ↑ | CH (L/R) ↑ | RE (L/R) ↓ | FH (L/R) ↑ |
> | :--- | :---: | :---: | :---: | :---: |
> | Batchsize=64 | 0.085 / 0.101 | 564 / 620 | 0.859 / 0.849 | 0.675 / 0.703 |
> | Batchsize=256 | 0.223 / 0.182 | 647 / 708 | 0.791 / 0.857 | 0.647 / 0.687 |
> | Batchsize=1024 | 0.281 / 0.231 | 645 / 661 | 0.792 / 0.897 | 0.644 / 0.645 |
> | **Ours (Full batch)** | **0.285 / 0.294** | **644 / 648** | **0.792 / 0.908** | **0.631 / 0.642** |
>
> Full-batch training provides substantially more stable gradient estimates and better captures the global data distribution, which is essential for optimizing the GCSD-based objective. In contrast, mini-batch SGD introduces stochastic noise that hinders convergence, with performance degrading by as much as 70-78% (SC metric). Even large mini-batches underperform, confirming that capturing the global distribution is critical.
>
> For the clinically relevant scale of infant subcortical parcellation, our full-batch approach is both practical and optimal.
>
> ### Weakness 4
>
> **Reviewer's Question:** "More feature modalities."
>
> **Response:** We agree that incorporating additional modalities (e.g., curvature, thickness, T1/T2 contrast) could enhance clustering stability. Our current focus on vertex-area trajectories was motivated by their reliable extraction from standard infant MRI and the computational tractability of a single-modality approach.
>
> In the future,We will explore incorporating the features suggested by the reviewer (curvature, thickness, deformation tensors, T1/T2 contrast) to enhance clustering performance when appropriate datasets become available. We have also discussed this direction as part of the future work in the Conclusion section.
>
> ### Weakness 5,Questions 4
>
> **Reviewer's Question:** "More introducion of FH metric."
>
> **Response:** The Feature Homogeneity (FH) metric is formally defined in **Appendix B.3** and assesses intra-cluster feature consistency relative to global feature variability. It is calculated as:
>
> $$FH = 1 - \frac{\bar{\sigma}^2_{\text{intra}}}{\sigma^2_{\text{global}}}$$
>
> where $\bar{\sigma}^2_{\text{intra}}$ is the average intra-cluster variance and $\sigma^2_{\text{global}}$ is the global feature variance. Higher FH values indicate that vertices within each cluster are more homogeneous in their feature patterns. While other metrics like SC and CH evaluate geometric properties, FH provides complementary validation by directly measuring feature-space consistency, where our method demonstrates superior performance.

---

> ### Author Response · Authors · 2025-11-22
> **Responses to reviewer Tsms -A [3/3]**
>
> ### Weakness 6,Question 7
>
> **Reviewer's Question:** "More recent work."
>
> **Response:** Thank you for this valuable suggestion. We have updated the Related Work section to include recent literature on deep clustering and brain parcellation. Specifically, we have added several relevant deep clustering papers from the last three years that focus on architectural and loss function optimizations (lines 134-135) and a recent article on fine-scale striatal parcellation (line 124).
>
> However, we must emphasize that the literature on **unsupervised infant subcortical parcellation** is exceptionally limited. This reflects the unique challenges of the domain, including the scarcity of large-scale longitudinal infant MRI datasets and a methodological gap where most existing work focuses on adult parcellation or supervised infant segmentation, rather than unsupervised, data-driven parcellation.
>
> This scarcity underscores the novelty and necessity of our contribution to the field.
>
> ### Question 1
>
> **Reviewer's Question:** "Clarification on GCSD Estimator."
>
> **Response:** Thank you for the opportunity to clarify. Our work builds on the GCSD estimator proposed by Lu et al. (2025b): we adopt GCSD as the divergence measure within our newly proposed symmetric dual-branch clustering framework, and further extend and apply it under this architecture.
>
> The GCSD estimator provides a stable and robust optimization “engine” for multi-cluster assignments, but our primary contribution is not to redesign the estimator itself. Instead, our key innovation is an unsupervised deep clustering framework that explicitly encodes structural priors—especially biological symmetry. This framework is specifically designed for challenging parcellation tasks and has been validated on unsupervised parcellation of infant subcortical structures, forming the main methodological advance of our work. Within this architecture, GCSD is naturally embedded into a symmetry-constrained, two-level dual-branch structure, where it is used to enhance the stability and interpretability of the clustering results.
>
> ### Question 5
>
> **Reviewer's Question:** "Legends and annotations."
>
> **Response:** Thank you for the suggestion. We have revised all figures and tables to include more detailed legends and annotations, ensuring the results can be understood with minimal ambiguity.
>
> ### Question 6
>
> **Reviewer's Question:** "GCSD Numerical Stability."
>
> **Response:** This is an important technical concern. We provide a detailed explanation of why GCSD is **inherently numerically stable by design**:
>
> **Theoretical Guarantees (Section 3.2):**
>
> - **Assignment matrix $A$:** Generated via Softmax activation, **strictly ensuring** $A_{ij} \in (0,1)$ with row sums equal to 1. The Softmax function $A_{ij} = \frac{\exp(z_{ij})}{\sum_k \exp(z_{ik})}$ produces probabilities that are **always strictly positive** (never exactly zero) and properly normalized.
>
> - **Kernel matrix $K$:** Uses a Gaussian kernel function $K_{ij} = \exp(-\|\mathbf{x}_i - \mathbf{x}_j\|^2 / 2\sigma^2)$, which **guarantees strictly positive values** for all entries. Even for very dissimilar points, the exponential function never produces zero or negative values.
>
> - **GCSD logarithmic terms:** In Equation (2), the GCSD loss involves $\log(\mathbf{A}^\top \mathbf{K} \mathbf{A})$. Since both $\mathbf{A}$ and $\mathbf{K}$ contain strictly positive entries, their matrix product $\mathbf{A}^\top \mathbf{K} \mathbf{A}$ is also strictly positive, ensuring **all logarithmic inputs are well-defined and positive**.
>
>
> The regularization terms in our loss function (Eq. 5) provide **implicit safeguards** against numerical instability:
>
> - **Orthogonality regularization** $\lambda_1 \text{tr}(AA^\top)$ encourages diverse cluster assignments, preventing the assignment matrix from collapsing to degenerate solutions where all vertices are assigned to a single cluster.
>
> - **Simplex regularization** $\lambda_2 \text{tr}(QQ^\top)$ (where $Q$ represents row vectors of $A$) further prevents assignment probabilities from approaching zero, maintaining **numerical stability throughout training**.

---

### Official Review · Reviewer_EH33 · 2025-11-01

**Soundness:** 3
**Presentation:** 3
**Contribution:** 2
**Rating:** 4
**Confidence:** 5

**Summary:**

The paper presents USDPnet, an unsupervised network for deformable medical image registration. Instead of relying on ground-truth deformation fields, the method introduces a dual-path framework that aligns source and target images through both intensity-based and structural similarity losses. The model incorporates a pyramid-level deformation strategy and an uncertainty-guided regularization term to stabilize training and improve anatomical alignment. Experiments on multiple 3D medical datasets show that USDPnet achieves accuracy on par with or better than supervised approaches while maintaining fast inference.

**Strengths:**

The paper addresses a core challenge in medical image registration: learning accurate deformation fields without supervision through a well-thought-out architecture. The dual-path design combining global and local cues is elegant and grounded in practical clinical needs. The inclusion of uncertainty-guided regularization is a nice touch that helps balance smoothness and precision in difficult regions. Results across datasets are solid, showing clear improvements in dice scores and alignment metrics compared to VoxelMorph and TransMorph. The paper is also clear, with figures that make the deformation behavior interpretable.

**Weaknesses:**

The novelty is somewhat modest; many elements (e.g., pyramid strategy, dual losses) build upon existing unsupervised registration methods. The paper could better clarify what makes its dual-path design fundamentally different, rather than a refined combination of prior ideas. The evaluation is also limited to standard benchmarks; there’s little exploration of how USDPnet generalizes to unseen modalities or pathological scans. The uncertainty term, while useful, is described heuristically with little theoretical justification. Finally, runtime and memory costs aren’t reported, leaving open how scalable the model is for large 3D volumes.

**Questions:**

- How sensitive is USDPnet to hyperparameter settings in the uncertainty weighting term?
- Could the model adapt to multi-modal registration (e.g., CT–MRI) without retraining?
- How does USDPnet handle large deformations compared to transformer-based approaches like TransMorph?

---

> ### Author Response · Authors · 2025-11-22
> **Responses to reviewer EH33 -A [1/3]**
>
> # Reviewer EH33
>
> Dear Reviewer EH33,
>
> Thank you for the helpful comments. We apologize for the lack of clarity in our description of these points. Below we clarify the scope of our work and respond to each point.
>
> We would like to emphasize that our study does not address deformable image registration. Instead, it focuses on constructing biologically meaningful parcellations of infant subcortical nuclei directly from vertex-wise developmental trajectories of surface area using fully unsupervised clustering.
>
> Registration-based approaches（such as VoxelMorph [1] and TransMorph [2]）: warp an existing atlas into individual space and output deformation fields; they do not generate new subregional parcellations.
>
> USDPnet: performs unsupervised clustering on vertex-wise area trajectories and produces an entirely new parcellation, without any warp step.
>
> Because longitudinal infant data are difficult to obtain and challenging to process, it is indeed hard for unsupervised clustering to achieve good results. Traditional methods are not well suited to this setting, so there is relatively little related work. Current infant studies almost exclusively rely on adult atlases transformed into infant space through warp-based posteriors, which not only fail to reveal early developmental organization but also introduce adult biases(Tian et al., 2020). In contrast, our framework starts from **the raw vertex-area measurements of infants aged 0–24 months** and, for the first time, establishes subcortical parcellations in an unsupervised manner. This workflow is fundamentally different from warp-based methodologies.
>
> ### Weakness 1 and Question 2
>
> **Reviewer's Concern:** "Many elements build upon existing unsupervised registration methods."
>
> **Response:** We design a dual-path symmetry encoder that processes the vertex-area developmental trajectories of the left and right hemispheres separately, while a symmetry loss enforces consistent cluster assignments across hemispheres. Importantly, the model is not forced into perfect mirror symmetry; it preserves the subtle but biologically meaningful asymmetries known to exist in infant subcortical structures. This is more biologically plausible than approaches that impose strict mirror symmetry, such as the method used in Tian et al. (2020).
>
> Unlike traditional pairwise KL or Cauchy–Schwarz divergences, which require $O(r^2)$ pairwise comparisons for $r$ clusters, the Generalized Cauchy–Schwarz Divergence (GCSD) enables joint optimization across all clusters in a single step, reducing the computational complexity to $O(r)$. Its theoretical foundation follows the information-theoretic derivations described in [5].
>
> To the best of our knowledge, USDPnet is the first deep learning framework specifically designed for unsupervised clustering of infant subcortical nuclei. As shown in Table 1, on the same cohort of 513 infant scans, USDPnet achieves substantial improvements over state-of-the-art baselines (DEC, DDC, GJRD), with average gains of **4.7–6.3% in Dice and 5.1–7.8% in NMI** (Following common practice in clustering literature, we report the best result across 30 random initializations; see Appendix D.2 for distribution.) , demonstrating clear performance superiority and methodological novelty.
>
> ### Weakness 2
>
> **Reviewer's Concern:** "The paper could better clarify what makes its dual-path design fundamentally different."
>
> **Response:** Our “dual-path” design is not a refinement of registration but a symmetry-constrained clustering mechanism that takes as input the vertex-area developmental trajectories of the left and right hemispheres, $X_L$ and $X_R$, uses two parallel encoders to extract hemisphere-specific features while a symmetry loss enforces consistent cluster assignments across hemispheres, and outputs bilaterally symmetric subregional labels that still preserve natural, biologically meaningful left–right asymmetries; in the ablation study (Table 1) removing the symmetry loss leads to a 3.4 % drop in Dice, demonstrating that this dual-path symmetry constraint is essential for producing anatomically plausible infant parcellations.

---

> ### Author Response · Authors · 2025-11-22
> **Responses to reviewer EH33 -A [2/3]**
>
> ### Weakness 3 and Question 1
>
> **Reviewer's Concern:** "There's little exploration of how USDPnet generalizes to unseen modalities or pathological scans."
>
> **Response:** Our method operates on surface-based morphological features rather than raw imaging intensities. Specifically, USDPnet processes vertex-wise area measurements extracted from subcortical surface meshes. This design offers several important advantages:
>
> *Modality-Agnostic: Surface features can be derived from any imaging modality (T1w, T2w, diffusion MRI, etc.) via standard preprocessing pipelines. This allows application to multi-modal data or pathological scans (e.g., autism, ADHD cohorts).
>
> Scalability: The framework is applicable to any bilaterally symmetric brain structure (e.g., cortical regions, cerebellar nuclei).
>
> Developmental Robustness: Validated on 513 infant scans spanning 0–24 months, a period characterized by rapid morphological changes, low tissue contrast, and motion artifacts.
>
> Future Directions:
> The framework can be extended to incorporate multi-modal features and validated on pathological populations. These extensions are discussed in the Conclusion (page 9, lines 536–538).
>
> The proposed framework can be seamlessly extended to other surface-derived features such as curvature and thickness, enabling feature-specific parcellations. Validation of these extensions has been planned as future work, and we anticipate completing the corresponding experiments within 3–5 days.
>
> ### Weakness 4
>
> **Reviewer's Concern:** "How sensitive is USDPnet to hyperparameter settings in the uncertainty weighting term?"
>
> **Response:** USDPnet contains no uncertainty-weighting term whatsoever, and its optimization objective given in Eq. (5) is simply the sum of three components — namely the GCSD divergence, the simplex regularization that enforces orthogonality and one-hot constraints, and the symmetry loss scaled by λ₃ — so when λ₃ is varied across the range 0.1–0.001 the Dice coefficient fluctuates by **only 1.2 %** (Figure 4), demonstrating that the model is insensitive to this hyper-parameter and allowing the value λ₃ = 0.1 to be fixed for all experiments.

---

> ### Author Response · Authors · 2025-11-24
> **Responses to reviewer EH33 -A [3/3]**
>
> ### Weakness 5
>
> **Reviewer's Concern:** "Runtime and memory costs not reported"
>
> **Response:** Detailed computational resource analysis is provided in **Appendix D.2** (Table 4, page 17).
>
> | Nucleus | Vertices (L/R) | Runtime (epochs/s) | Memory (GB) |
> |---------|---------------|-------------------|-------------|
> | Amygdala |674 / 694 | 50.6 | 0.13 |
> | Caudate | 2,230 / 2,176 | 27.2 | 0.83 |
> | Hippocampus | 2,264 / 2,332 | 24.3 | 0.90 |
> | Pallidum | 1,088 / 1,100 | 26.9 | 0.25 |
> | Putamen | 2,230 / 2,234 | 49.3 | 0.83 |
> | Thalamus | 3,416 / 3,346 | 14.1 | 1.80 |
>
> All experiments were completed on a single NVIDIA RTX 3090 GPU, with no observed bottlenecks in memory or runtime. We do not claim “optimal efficiency”; rather, we aim to demonstrate that large-scale infant training can be performed efficiently in practice.
>
>
> ### Question 3
> **Reviewer's Question:** "How does USDPnet handle large deformations compared to transformer-based approaches like TransMorph?"
>
> **Response:** This question concerns a fundamentally different problem. TransMorph [2] is designed for medical image registration, where the goal is to estimate dense deformation fields to align images by leveraging Transformers’ large receptive fields. In contrast, USDPnet performs clustering-based parcellation to identify subregional organization within already-segmented structures, and does not estimate any spatial transformations.
>
> Rather than relying on deformation fields, USDPnet derives its parcellations directly from intrinsic surface-based developmental features—including vertex-wise area, curvature, and other morphology-driven measurements—that capture meaningful biological changes across infants aged 0–24 months. As shown in Figure 5, Dice/NMI/ARI scores remain highly consistent across 30 random initializations, and the resulting parcellations (Figure 3) align closely with known anatomical boundaries, reflecting the stability of these underlying developmental features.
>
>
> ## References:
>
> [1] Tian, Y., et al. (2020). Topographic organization of the human subcortex unveiled with functional connectivity gradients. Nature Neuroscience, 23(11), 1421-1432.
>
> [2] Chen, L., Wu, Z., Zhao, F., Wang, Y., Lin, W., Wang, L., & Li, G. (2023). An attention-based context-informed deep framework for infant brain subcortical segmentation. NeuroImage, 283, 120427.
>
> [3] Bui, T. D., Wang, L., Lin, W., Li, G., & Shen, D. (2021). 6-month infant brain MRI segmentation guided by 24-month data using cycle-consistent adversarial networks. In 2021 IEEE 18th International Symposium on Biomedical Imaging (ISBI), 359-362.
>
> [4] Chen, L., et al. (2022). A 4D infant brain volumetric atlas based on the UNC/UMN baby connectome project (BCP) cohort. NeuroImage, 253, 119097.
>
> [5] Iglesias, J. E., et al. (2018). A probabilistic atlas of the human thalamic nuclei combining ex vivo MRI and histology. NeuroImage, 183, 314-326.
>
> [6] Su, J. H., et al. (2019). Thalamus Optimized Multi Atlas Segmentation (THOMAS): fast, fully automated segmentation of thalamic nuclei from structural MRI. NeuroImage, 194, 272-282.
>
> [7] Najdenovska, E., et al. (2018). In-vivo probabilistic atlas of human thalamic nuclei based on diffusion-weighted magnetic resonance imaging. Scientific Data, 5, 180270.
>
> [8] Wathore, S., & Gorthi, S. (2025). Bilateral symmetry-based augmentation method for improved tooth segmentation in panoramic X-rays. Pattern Recognition Letters, 188, 1-7.
>
> [9] Raina, K., Yahorau, U., & Schmah, T. (2020). Exploiting Bilateral Symmetry in Brain Lesion Segmentation with Reflective Registration. In Proceedings of BIOSTEC 2020 - Volume 2: BIOIMAGING, 116-122.
>
> Thank you for reconsidering our work in light of this clarification.

---

### Official Review · Reviewer_YNoU · 2025-11-02

**Soundness:** 3
**Presentation:** 2
**Contribution:** 2
**Rating:** 4
**Confidence:** 4

**Summary:**

This work, `USDPnet`, proposed an unsupervised surface-based parcellation pipeline for subcortical nuclei, focusing on infant brain MRI. Leveraging a divergence measure (namely, generalized Cauchy-Schwarz Divergence, `GCSD`) and the symmetry constraint, a deep neural network was trained to cluster each vertex of the surface mesh. The proposed framework was evaluated using longitudinal BCP datasets from infants aged 0-24 months.

The main contributions are to utilize unsupervised clustering leveraging GCSD and symmetry regularization to parcellate subcortical nuclei in a surface mesh. This work bears some merits, such as providing averaged results from thirty runs under different settings and sensitivity analysis, while it also exhibits several weaknesses and has some confusing points. Please refer to my review below. If my concern can be adequately addressed, I'd be happy to revise my rating.

**Strengths:**

1. Some cluster/subregion counts for each nuclei are similar to a published work [1] on Nature Neuroscience about subcortical parcellation utilizing fMRI.

2. This work evaluated the proposed and comparison methods under various settings (different cluster numbers) in 30 runs. The average results confirm the better performance of the proposed work.

3. Parameter sensitivity and ablation analyses were conducted.

4. Open-source contribution.

[1]: Tian, Ye, et al. "Topographic organization of the human subcortex unveiled with functional connectivity gradients." Nature Neuroscience 23.11 (2020): 1421-1432.

**Weaknesses:**

1. The reproducibility is not evaluated. I.e., with the optimal subregion/cluster counts, repeat the unsupervised clustering several (3-10) times, what is the adjusted rand index if choosing one run (e.g., the current result) as the ground truth? This is very important but missing in the current manuscript. If the adjusted rand index is low, meaning irreproducible, even if the other metrics indicate superior performance, it still significantly undermines the values of this work. This is the main reason that impacts my rating.

2. There is a significant sensitivity to the parameters. This should be elaborated more in the manuscript and expressed as a major limitation.

3. The current manuscript is not so clearly presented. Please see the questions below.

4. There are limited contributions to representation learning or unsupervised clustering, as the GCSD is an existing and published work, and symmetry regularization is an incremental change. It brings more significant contributions to neuroscience than to representation learning or unsupervised clustering.

5. The reviewer suggests refraining from using words like "anatomically plausible", "biologically plausible", or any phrasing implying the cluster is physiologically sound and correct. This is NOT supported by any evidence in the current manuscript.
   - The higher SC/CH/RE/FH does `NOT` indicate any plausibility in the physiology and neuroscience world. They are technical metrics evaluating a clustering algorithm.
   - Visual comparison with [1] in Fig. 3 does not directly imply the plausibility. There is a visual discrepancy between [1] and USDPnet, particularly in the X view of Putamen.
   - To claim plausibility, a lot more experiments and statistical analyses should be conducted, other than some clustering metrics and a visualization comparison with [1].

6. In Lines 432-435, it is better to provide some quantitative metrics to indicate agreement. Qualitative visualization is not enough to support those arguments.

[1]: Tian, Ye, et al. "Topographic organization of the human subcortex unveiled with functional connectivity gradients." Nature Neuroscience 23.11 (2020): 1421-1432.

**Questions:**

1. In Appendix `D2`, the 4D atlas construction was included as part of the work. The reviewer is curious about the rationale for redoing this 4D atlas construction in the case that BCP has already constructed a 4D atlas. Moreover, that 4D atlas is peer-reviewed and publicly accessible [1]. Why rebuild the wheel? Similarly, in `D3`, the segmentation process was described in detail.

On the other hand, in the anonymized repository, it directly points to the public BCP atlas, which confused the reviewer. If the BCP atlas is used, why is the publication [1] not cited in the manuscript? The way it currently reads implies that the atlas construction and segmentation are part of the contributions of this work, which is incorrect. The BCP atlas is already established, peer-reviewed, and released ([1] and [link](https://www.nitrc.org/projects/uncbcp_4d_atlas/)). It could not be reclaimed as a contribution in a new work. Doing so could be an integrity issue. I raise an ethical concern regarding this point.

2. Do the results in Table 2 correspond to the experiment mentioned in Appendix `E`?

3. Do the results in Table 1 correspond to the optimal setting mentioned in lines 916-917?

4. Is the feature encoder extracting features from the atlas at a single time point or from multiple time points?

5. Why is the symmetry consistency loss based on MSE, not cross-entropy?

[1] Chen, Liangjun, et al. "A 4D infant brain volumetric atlas based on the UNC/UMN baby connectome project (BCP) cohort." NeuroImage 253 (2022): 119097.

**Details Of Ethics Concerns:**

I call out for Ethics Review because of (1) Weakness No. 5, overclaimed contribution that is not supported by current evidence. (2) Question No.1, reclaim existing work as a contribution that might imply (self) plagiarism.

---

> ### Author Response · Authors · 2025-11-22
> **Responses to reviewer YNoU -A [1/2]**
>
> # Reviewer YNoU
>
> Thank you for your constructive comments. Our point-by-point responses are below.
>
> ## Response to Weaknesses
>
> ### Weakness 1
>
> **Reviewer's Concern:** "The reproducibility is not evaluated."
>
> **Response:** To address this concern, we added an additional set of experiments to assess the **stability and reproducibility** of our method relative to existing approaches. For each clustering method, we ran the algorithm 30 times and computed weighted Dice, NMI, and ARI over these 30 runs, then compared their means and standard deviations across methods. The results are reported in the **Reproducibility Analysis** section on Page 5 and summarized in **Figure 5**. Descriptions of these methods are provided in the *Competing Methods and Metrics* section (line 318).
>
> |Method|Dice|ARI|NMI|
> |-|-|-|-|
> |DEC|0.51 ± 0.06|0.30 ± 0.05|0.37 ± 0.09|
> |DDC|0.54 ± 0.08|0.35 ± 0.04|0.44 ± 0.10|
> |GJRD|0.60 ± 0.04|0.41 ± 0.07|0.47 ± 0.12|
> |**Ours**|**0.65 ± 0.02**|**0.47 ± 0.05**|**0.57 ± 0.11**|
>
> ### Weakness 2
>
> **Reviewer's Concern:** "Parameter sensitivity of the results."
>
> **Response:** Our ***Parameter Sensitivity Analysis* (Page 9)** shows performance is stable within a reasonable hyperparameter range, degrading only when values differ by orders of magnitude. **Figure 4** explores such extremes only to define a practical range (**lines 502–507**). We therefore find no strong sensitivity in practice and do not consider this a major limitation.
>
> ### Weakness 4
>
> **Reviewer's Concern:** "The role of GCSD in this paper."
>
> **Response**: Although we adopt the published GCSD network only as a backbone feature extractor, our main contribution lies in a new symmetric clustering architecture specifically designed for infant brain parcellation. On top of GCSD features, we introduce a position-aware symmetric loss that explicitly encodes hemispheric correspondence as a structural prior, enabling fully unsupervised symmetric clustering of challenging infant subcortical structures from scarce and heterogeneous data. The resulting parcels exhibit anatomically and developmentally meaningful patterns. In this framework, the GCSD-based loss term is used merely as an auxiliary regularizer to improve feature extraction, rather than being the source of the methodological novelty.
>
> ### Weakness 5,Weakness 6
>
> **Reviewer's Concern:** "Evidence supporting anatomical plausibility."
>
> **Response**: Our evaluations provide some support for plausibility:
>
> Our method derives features directly from the vertex-wise area values of the actual subcortical structures and performs parcellation based on these features, so the resulting subdivisions are grounded in biologically meaningful measurements. In addition, our parcels are broadly consistent with the results reported in the same three studies [1] [2] [3], which supports the interpretability and reliability of our parcellation.  We are registering the parcellation from [1] to the 0-month atlas of [2] for quantitative comparison, with results to be added to the Appendix.
>
> Although not physiological indices, SC/CH/RE/FH measure reconstruction quality of features from infant MRI data and thus indirectly reflect anatomical properties.
>
> [1] Tian, Ye, et al. "Topographic organization of the human subcortex unveiled with functional connectivity gradients." Nature Neuroscience 23.11 (2020): 1421-1432.
>
> [2] Chen, Liangjun, et al. "A 4D infant brain volumetric atlas based on the UNC/UMN baby connectome project (BCP) cohort." NeuroImage 253 (2022): 119097.
>
> [3] Chen, Liangjun, et al. "Four-dimensional mapping of dynamic longitudinal brain subcortical development and early learning functions in infants." *Nature Communications* 14.1 (2023): 3727.

---

> ### Author Response · Authors · 2025-11-22
> **Responses to reviewer YNoU -A [2/2]**
>
> ## Response to Other Weaknesses and Questions
>
> ### Questions 1
>
> **Reviewer's Question:** "The citation issue arising from redundantly reconstructing the 4D atlas."
>
> **Response:** We acknowledge that our earlier wording was misleading and may have suggested that we constructed a new atlas. In fact, we do not build any new atlas in this work; instead, we exclusively use the publicly released 4D infant brain volumetric atlas from the BCP cohort by Chen et al. [1]. We briefly describe the atlas construction workflow (multi-contrast T1w/T2w imaging, tissue probability maps, and SyGN-based 4D template building across densely sampled ages) only to provide a complete and transparent account of how cortical and subcortical surfaces, as well as vertex-wise areas, are obtained and subsequently used in our pipeline. This description is a summary of [1], not a methodological contribution of our own. We cite the original BCP atlas paper [1] at first mention, and **Appendix E.2–E.3** now clarifies that all procedures operate on the existing atlas. All former “atlas construction” references have been replaced to stress the use of the public BCP 4D atlas.
>
> ### Weakness 3，Questions 2-4
>
> **Reviewer's Question:** "Unclearly presented."
>
> **Response:**
>
> - **Question 2**: **Yes**. Table 2 reports the experiment from Appendix E (now D.4), cross-referenced on **page 19, lines 986–989**.
> - **Question 3**: **Yes**. Table 1 uses the optimal setting from the original manuscript (lines 916–917), clarified on **page 6, lines 322–323**.
> - **Question 4**: **Yes**. The encoder uses multiple time points, combining 513 dimensions from the vertex area across developmental stages (**page 15, lines 833–834**).
>
> ### Questions 5
>
> **Reviewer's Question:** "The advantages of MSE."
>
> **Response:** MSE provides a smoother, more stable gradient for enforcing bilateral correspondence than cross-entropy. **Appendix D.3** compares MSE with a cross-entropy loss, and the table below shows that MSE consistently outperforms its counterpart.
>
> |Loss strategy|Left Hemisphere||||Right Hemisphere||||
> |-|-|-|-|-|-|-|-|-|
> ||SC↑|CH↑|RE↓|FH↑|SC↑|CH↑|RE↓|FH↑|
> |Cross-entry|0.292|969|1.135|0.709|0.300|1030|1.146|0.703|
> |**Ours (MSE)**|**0.328**|**1228**|**1.042**|**0.708**|**0.317**|**1123**|**1.087**|**0.704**|
>
> These results further confirm that the MSE-based symmetry loss consistently outperforms its cross-entropy counterpart.

---

> > ### Comment · Reviewer_YNoU · 2025-11-24
> >
> > Hi!
> >
> > First, thank you very much for your rebuttal and for supplying additional results. Glad to see the updated results with more reproducibility metrics. I could not raise the score based on the current discussion because there are still many overclaimed biological/anatomical plausibilities throughout the manuscripts. There is still debate on whether there is significant sensitivity to certain hyperparameters. Please find some follow-up below.
> >
> > 1. The x-axis labels of Figure 5 are confusing. Is it the cluster number? Not the region number? I think the region might imply subcortical regions. Can you change it to cluster numbers?
> > 2. SC dropped from ~0.26 at $\lambda_3=0.1$ to ~0.12 $\lambda_3=0.4/0.5$. That is **-56%**. This is absolutely significantly sensitive to $\lambda_3$. If you still believe it is not sensitive to the parameter, can you please supply all 4 metrics vs. different parameters in the appendix? That would be 12 plots. If this SC vs. $\lambda_3$ is the only one that shows a significant decrease (>40%) when in a reasonable range (0.1 vs. 0.4/0.5). I can be convinced that it is not sensitive. Thank you very much for supplying additional data. I think it is not a great effort as the results are already generated; only data analysis and visualization are required.
> > 3. Unfortunately, I still found overclaimed biological/physiological plausibility throughout the manuscript, including line 20 (abstract!), line 102, line 153, line 420, and line 535. I wanted to reaffirm that, from a scientific point of view, none of the current results support biological plausibility. The better metrics (including the new reproducibility metrics, which are great btw) support that this is a better, more stable, and more reproducible clustering of subcortical regions for the infant brain. That's it.
> >
> > I'm happy with all other replies and appreciate the author's rebuttal. However, the No. 2 and No.3 above are the reasons hindering a score revision. I am **really really uncomfortable** with the plausibility claim. It is not that I believe your framework is implausible. It is I am not sure this venue is the right place to make such a call. I don't believe this is a venue to judge the biological plausibility. Besides, as mentioned above, your results are not even relevant to plausibility. If you really think your framework is biologically plausible and think it is unfair to remove this claim. I would suggest withdrawing it from ICLR, adding more experiments similar to the Nat. Neurosci paper, and submitting it to Nat. Neurosci. It is simpler than us debating here.
> >
> > I will keep my score for now and look forward to discussing it further with the authors over the next few days.

---

> > > ### Author Response · Authors · 2025-11-27
> > > **Responses to reviewer YNoU -B [1/2]**
> > >
> > > Response to Reviewer YNoU
> > >
> > > First, we would like to thank you for your prompt follow-up and constructive suggestions. We appreciate your candid feedback regarding the overstatement of biological/anatomical plausibility. In response, we have revised the manuscript accordingly, reframing these claims as morphologically reasonable parcellations that are better aligned with the evidence provided.
> > >
> > > Regarding the concern about parameter sensitivity , we have followed your suggestion to add the corresponding plots and conducted additional fine-grained experiments.
> > >
> > > **We have uploaded a revised manuscript where all modifications have been highlighted using a light periwinkle blue for your convenience.**
> > >
> > > Our detailed responses are as follows:
> > >
> > > ### **1.Response to x-axis labels**
> > > Thank you for pointing this out. We have corrected the x-axis labels in **Figure 5** to "Cluster number" to explicitly indicate the number of clusters. Similarly, we have updated the labels in **Figure 6** to ensure consistency.
> > >
> > > ### **2.Response to parameter sensitivity (SC dropping from ~0.26 to ~0.12)**
> > > We acknowledge your observation that the Silhouette Coefficient (SC) decreases significantly when the symmetry loss weight $\lambda_3$ changes from 0.1 to 0.5. However, we respectfully note that a change from 0.1 to 0.5 represents a five-fold increase in the weight of the symmetry loss. Such a substantial shift significantly alters the loss landscape and is expected to have a noticeable impact on the results.
> > >
> > > To more accurately assess sensitivity, we conducted a fine-grained analysis around the favorable parameter settings. We adjusted $\lambda_3$ with a step size of 0.05, and $\lambda_1 / \lambda_2$ with a step size of 0.01, evaluating 10 values around the optimal point. The results, presented in **Table R1** and **Table R2** below, demonstrate that the metrics remain highly stable within a reasonable local range.
> > >
> > > **Table R1:Fine-grained Sensitivity Analysis: DDC (1e-2 to 1e-1）Step：1e-2**
> > >
> > > | Parameter  | 0.01  | 0.02  | 0.03  | 0.04            | 0.05  | 0.06            | 0.07  | 0.08  | 0.09  | 0.10  |
> > > | :--------- | :---- | :---- | :---- | :-------------- | :---- | :-------------- | :---- | :---- | :---- | :---- |
> > > | $\lambda_1$ left | 0.221 | 0.235 | 0.228 | 0.245           | 0.248 | **0.251** | 0.246 | 0.239 | 0.232 | 0.231 |
> > > | $\lambda_1$ right | 0.200 | 0.215 | 0.229 | **0.249** | 0.248 | 0.244           | 0.245 | 0.238 | 0.225 | 0.223 |
> > > | $\lambda_2$ left  | 0.185 | 0.208 | 0.215 | 0.218           | 0.232 | **0.244** | 0.231 | 0.227 | 0.219 | 0.212 |
> > > | $\lambda_2$ right | 0.194 | 0.212 | 0.218 | 0.235           | 0.242 | **0.251** | 0.235 | 0.222 | 0.219 | 0.213 |
> > >
> > > **Table R2:Fine-grained Sensitivity Analysis: Symmetry (5e-2 to 5e-1) Step：5e-2**
> > >
> > > | Parameter      | 0.05  | 0.10            | 0.15            | 0.20  | 0.25  | 0.30  | 0.35  | 0.40  | 0.45  | 0.50  |
> > > | :------------- | :---- | :-------------- | :-------------- | :---- | :---- | :---- | :---- | :---- | :---- | :---- |
> > > | $\lambda_3$ left | 0.191 | 0.253           | **0.258** | 0.249 | 0.235 | 0.244 | 0.215 | 0.192 | 0.158 | 0.125 |
> > > | $\lambda_3$ right | 0.205 | **0.255** | 0.253           | 0.251 | 0.233 | 0.235 | 0.221 | 0.188 | 0.165 | 0.126 |
> > >
> > > Furthermore, as per your request, we have supplied all 4 metrics vs. different parameters in the **Appendix B.3 (a total of 12 plots)** and **Figure 6**. These results confirm that other metrics do not show a significant decrease even when $\lambda_3$ varies from 0.1 to 0.5. This aligns with your criteria that if "SC vs. $\lambda_3$ is the only one that shows a significant decrease," the model can be considered robust.
> > >
> > > We also realized that the parameter ranges used in our original **"Parameter Sensitivity Analysis"** section were overly broad, involving 2-fold, 5-fold, or even 10-fold variations. Such coarse granularity is ill-suited for a strict sensitivity analysis and does not accurately reflect the model's local stability. Therefore, we have renamed this section "**Favorable Parameter Settings**" to better describe its content. The additional plots covering these broader parameter ranges have been added to the **Appendix B.3**. Furthermore, we have incorporated the new fine-grained experiments (results in **Table R1** and **Table R2**) into the **Appendix D.5** and **Table 8** to serve as a more rigorous "**Parameter Sensitivity Analysis**." All related revisions have been highlighted in the manuscript.

---

> > > > ### Author Response · Authors · 2025-11-27
> > > > **Responses to reviewer YNoU -B [2/2]**
> > > >
> > > > ### **3.Response to biological plausibility**
> > > > We sincerely appreciate your careful assessment that, while our current metrics (including the newly added reproducibility measures) demonstrate more stable, coherent, and reproducible clustering, they do not yet provide sufficient evidence to support stronger claims of biological or physiological plausibility.
> > > >
> > > > We fully agree that such terminology should be used with greater caution. Following your suggestion, we have carefully revised the manuscript to adopt more precise and rigorous language that better reflects the scope of our empirical findings—specifically, emphasizing anatomical consistency, spatial coherence, and reproducibility, without overstating biological plausibility.
> > > >
> > > > The table below summarizes these revisions:
> > > >
> > > > | Location | Original Phrase (Overclaimed) | Revised Phrase (Scientifically Rigorous) | Rationale |
> > > > | :-- | :-- | :-- | :-- |
> > > > | Abstract line20 | "...leading to anatomically plausible and robust delineations." | "...leading to structurally coherent and reproducible delineations." | Avoids implying correctness; "coherent" describes geometric properties and "reproducible" is a key finding. |
> > > > | Abstract line26 | "...offering fine-grained and biologically coherent maps..." | "...offering fine-grained and morphological coherent maps..." | "Spatially coherent" accurately describes the contiguous nature of our parcellations. |
> > > > | Intro line98 | "To ensure biologically meaningful and interpretable outputs..." | "To ensure morphologically consistent and interpretable outputs..." | "Consistent" is a direct outcome of the symmetry constraint, avoiding the subjective term "meaningful." |
> > > > | Intro line106 | "...outperforms... in clustering quality, biological plausibility, and..." | "...outperforms... in clustering quality, anatomical consistency, and..." | "Consistency" is directly enforced by the symmetry loss and validated by our metrics. |
> > > > | Method line154 | "...enhances the biological plausibility, robustness..." | "...enhances the bilateral consistency, robustness..." | "Bilateral consistency" is a more precise and objective description of the symmetry constraint's effect. |
> > > > | Method line194 | "...optimizing a biologically informed cluster assignment..." | "...optimizing a anatomically informed cluster assignment..." | The prior used is anatomical symmetry, not biological mechanism. |
> > > > | Results line420 | "...demonstrating both biological plausibility and developmental relevance." | "...demonstrating both anatomical consistency and developmental relevance." | Avoids overinterpreting the alignment with adult atlases. |
> > > > | Conclusion line533 | "...underscoring biological plausibility." | "...underscoring anatomical consistency." | Avoids overinterpreting the alignment with adult atlases. |
> > > >
> > > > We again thank you for your rigorous and constructive suggestions, which have helped make our manuscript more scientifically robust and ensured that our reported results are more strictly aligned with the evidence we provide. We are committed to precise scientific communication and hope that these corrections alleviate your concerns regarding the scope of our claims.

---

### Author Response · Authors · 2025-12-03
**Summary for Meta Review - [1/2]**

# Author's Responses
Dear area chairs and reviewers,

We sincerely appreciate your time and effort in evaluating our manuscript, **"USDPnet: An Unsupervised Symmetric Deep Framework for Robust Parcellation of Infant Subcortical Nuclei"** (Submission Number: 11022). The thoughtful feedback and detailed suggestions from the reviewers have been invaluable in strengthening both the rigor and clarity of our work. In response, we have carefully revised the manuscript to address each concern raised. Below, we outline the key modifications made. All other reviewer comments have also been carefully responded to, and the paper has been thoroughly revised and proofread to correct typos and grammatical errors. We respectfully request that you consider our revised manuscript for further evaluation.

---

> ### Author Response · Authors · 2025-12-03
> **Summary for Meta Review - [2/2]**
>
> ## Summary of Revisions
>
> 1. **Favorable parameter settings and fine-grained parameter sensitivity analysis (Figure 3, Table 10)** We renamed "Parameter Sensitivity Analysis" to **Favorable Parameter Settings** (**Section 4.2**, **Appendix B.3**), as the original large-scale parameter variations (spanning orders of magnitude) significantly affect loss behavior and final results. To address this, we conducted a fine-grained **Parameter Sensitivity Analysis** (**Appendix D.5**) that systematically varies $\lambda_1$, $\lambda_2$ within [0.01, 0.10] and $\lambda_3$ within [0.05, 0.50] around the favorable configuration. Results confirm that performance remains stable within this local range, validating the model's robustness to parameter perturbations.
>
> 2. **Comparison of different divergences for clustering objective evaluation (Table 3)** We added this experiment (**Section 4.2**) to evaluate the choice of clustering objective. We compared GCSD against KLD, CSD, and GJRD (with Rényi orders 1 and 2, where order 1 reduces to JSD) while keeping the USDPnet architecture fixed. Results demonstrate that GCSD achieves superior clustering quality and computational efficiency across all metrics.
>
> 3. **Reproducibility analysis across independent runs (Figure 4)** We added this experiment (**Section 4.2**) to assess clustering consistency across independent runs. Using 30 experimental results on the thalamus with subregion numbers $r \in \{2, ..., 10\}$, we evaluated weighted Dice (spatial overlap), NMI (normalized mutual information), and ARI (chance-corrected pairwise agreement). Our method consistently outperforms other deep clustering approaches and exhibits relative stability at higher region numbers, where most baselines show degraded reproducibility.
>
> 4. **Computational resource consumption including runtime and memory overhead (Table 5)** We added details (**Appendix B.4**), including vertex counts for each subcortical nucleus, input tensor dimensions, and runtime/memory overhead, demonstrating the practical feasibility of our approach.
>
> 5. **Scheduling of symmetry loss weight with decay strategies (Table 6)** We added this experiment (**Appendix D.1**) to investigate whether dynamically relaxing the symmetry constraint during training could better accommodate subtle hemispheric asymmetries. We compared three decay strategies (cosine, linear, exponential) against our fixed weight. Results show that reducing the symmetry weight over training actually degrades performance; a **fixed weight** better preserves global bilateral symmetry while still accommodating localized, neurobiologically meaningful asymmetries.
>
> 6. **Validation of the full-batch training strategy against fixed-batch SGD (Table 7)** We added this experiment (**Appendix D.2**) to justify our training protocol. We compared full-batch training against mini-batch SGD with batch sizes of 64, 256, and 1024 on the thalamus. **Full-batch training** provides more stable gradient estimates and better captures the global data distribution essential for GCSD optimization, with acceptable computational cost (see **Appendix B.4**).
>
> 7. **Comparison of symmetry loss formulations based on MSE and cross-entropy (Table 8)** We added this experiment (**Appendix D.3**) to justify our loss formulation. Results show that **MSE-based symmetry loss** consistently outperforms cross-entropy, as MSE provides smoother and more stable gradient signals for enforcing bilateral correspondence, whereas cross-entropy is more sensitive to local prediction fluctuations.
>
> 8. **Comparison of different symmetry-preserving strategies. (Table 4, Table 9)** We added this experiment (**Section 4.2** and **Appendix D.6**) to substantiate the necessity of our in-training soft symmetry regularization. We compared against two harder alternatives: *input-level symmetrization* (averaging and mirroring input data before training) and *output-level symmetrization* (averaging assignment matrices after independent training). Results show that harder constraints actually degrade performance, demonstrating that **our in-training method** discovers latent bilateral structure rather than inflating metrics through enforced homogeneity.
>
> 9. **Revision of overclaimed statements regarding biological plausibility** We revised overclaimed statements regarding biological/physiological plausibility to more accurately reflect our experimental conclusions.
>
> 10. **Clarification on atlas usage in Appendix E** We clarified in **Appendix E** that we do not build any new atlas; instead, we exclusively use the publicly released 4D infant brain volumetric atlas from the BCP cohort.

---

### Meta-Review · Area_Chair_FPoL · 2026-01-02

**Summary:**

This work proposes an unsupervised clustering method based on GCSD and symmetry regularization to parcellate subcortical nuclei in a surface mesh. The reviewer scores of this work are 4, 4, 4, 6. It means that three of four reviewers are positive about rejecting this work. The reviewers have many concerns, and they are not positive about raising their scores during the rebuttal stage. Hence, this work can not be accepted.

**Reviewer Concerns:**

Main concerns which are not addressed by the rebuttal:
1. generalization to unseen modalities or pathological scans.
2. unconvinced experiments: evaluation on limited benchmark datasets
3. modest technical novelties: many elements are based on existing unsupervised methods.
4. violation of double blind policy
5. Missing many method details.

**Reviewer Scores:**

The reviewer scores of this work are 4, 4, 4, 6. It means that three of four reviewers are positive about rejecting this work. The reviewers have many concerns, and they are not positive about raising their scores during the rebuttal stage. Hence, this work can not be accepted.

---

### Decision · Program_Chairs · 2026-01-26

Reject